# A distinct assembly pathway of the human 39S late pre-mitoribosome

Jingdong Cheng 1✉, Otto Berninghausen 1 & Roland Beckmann 1✉

Assembly of the mitoribosome is largely enigmatic and involves numerous assembly factors. Little is known about their function and the architectural transitions of the pre-ribosomal intermediates. Here, we solve cryo-EM structures of the human 39S large subunit pre-ribosomes, representing five distinct late states. Besides the MALSU1 complex used as bait for affinity purification, we identify several assembly factors, including the DDX28 helicase, MRM3, GTPBP10 and the NSUN4-mTERF4 complex, all of which keep the 16S rRNA in immature conformations. The late transitions mainly involve rRNA domains IV and V, which form the central protuberance, the intersubunit side and the peptidyltransferase center of the 39S subunit. Unexpectedly, we find deacylated tRNA in the ribosomal E-site, suggesting a role in 39S assembly. Taken together, our study provides an architectural inventory of the distinct late assembly phase of the human 39S mitoribosome.

[1] Gene Center and Department for Biochemistry, LMU Munich, München, Germany. ✉email: jcheng@genzentrum.lmu.de; beckmann@genzentrum.lmu.de

The human mitochondrial 55S ribosome (mitoribosome) consists of a 28S small ribosomal subunit (mtSSU), formed by a 12S rRNA and 29 mitochondrial ribosomal proteins (MRPs), and the 39S large ribosomal subunit (mtLSU), formed by a 16S rRNA and 50 MRPs. It is specialized to synthesize 13 hydrophobic polypeptides, which are essential for oxidative phosphorylation (OXPHOS), thus playing an essential role in maintaining the mitochondrial function. Numerous mutations in mitochondrial rRNA, the ribosomal proteins or the translation regulators (initiation, elongation factors) are known to cause human diseases[1,2]. With the recent new development in cryo-electron microscopy (cryo-EM), we have gained wealth information about both the mtSSU and mtLSU structures, and also about the mode of translation by the active 55S ribosome[3–12]. Comparing with its bacterial ancestor, 39S large ribosomal subunits share most common structural units, such as six subdomains in the 16S rRNA (domain I to VI), the peptide transferase center (PTC), L1 and L7/12 stalks, and a central protuberance (CP). However, there are substantial differences in structure and composition, such as an increased number of mitochondrial ribosomal proteins and substantially different rRNA content[13], resulting in a high protein to RNA ratio in case of the human mitoribsome[14]. Moreover, a valine tRNA forms the central protuberance, thereby replacing the 5S rRNA[4]. Taken together, this suggests a deviating assembly process for the mitochondrial ribosomes.

Ribosome biogenesis in general involves the stepwise folding of the ribosomal RNA and recruitment of ribosomal proteins with the help of the assembly factors. Similar to the more extensively studied bacterial ribosome, the mitochondrial 39S large subunit requires dozens of assembly factors (AFs), such as methyltransferases[15–17], pseudouridine synthases[18,19], GTPases[20–25], RNA helicases[26,27], and other factors[28–31]. However, only a handful of homologs of these factors have been captured on bacterial ribosome assembly intermediates in structural studies[32]. Unlike the bacterial ribosome, until now there are only four highly-conserved modification sites mapped on the mtLSU: one pseudouridylation at U1397, which is carried out by RPUSD4[18], and three 2'-O-ribose methylations, which are catalyzed by three methyltransferases, MRM1, MRM2, and MRM3[15]. MRM1 modifies G1145 during the very early mitoribosome assembly pathway, since it appears to modify naked RNA and does not co-sediment with pre-mitoribosomal particles[15]. The other two sites, U1369 and G1370, are modified by MRM2 and MRM3, respectively, and are located on the A-loop in which these modifications are suggested to influence A site tRNA accommodation[17]. In addition, NSUN4 has been identified as an rRNA m5C methyltransferase for the methylation of 12S rRNA of the mtSSU on position C911. It belongs to the same protein family as the bacterial assembly factor RsmB, however, in contrast to RsmB it cannot directly bind to RNA without the help of mTERF4. mTERF4 belongs to the highly-conserved family of mitochondrial transcription termination factors (mTERF), which usually have nucleic acid binding activity. Notably, the NSUN4–mTERF4 complex has also been involved in ribosomal subunit joining in mitochondria[16,29,33,34].

Recently, structures of two late assembly intermediates of the human 39S mitoribosome were reported, in which one of them has already adopted the mature 16S rRNA conformation[31]. Here, the MALSU1 complex (MALSU1, L0R8F8, and mt-ACP) was identified in both intermediates. MALSU1 belongs to the RsfS protein family which, similar to eIF6 in Eukaryotes, binds to uL14 on the large subunit and serves as an anti-association factor preventing premature subunit joining in bacteria[30,35]. Moreover, several large subunit assembly intermediates of kinetoplastid LSU mitoribosome have also been solved by cryo-EM recently[36–38]. Although, we share some of the common AFs with kinetoplastids, the majority of their AFs are species specific and do not exist in human mitochondria. Since mitoribosome assembly appears to be quite diverse between species, the pathway of human 39S mitoribosome assembly remains largely enigmatic.

In this work, we use tagged assembly factors MALSU1 and GTPBP10 as baits for affinity purification to isolate late assembly intermediates of the 39S LSU mitoribosome. Subsequent cryo-EM single-particle analysis reveals the structures of nine late assembly intermediates with distinct AF compositions and arrangements. We observe an overall conserved the rRNA maturation pathway and several assembly factors apparently functioning to delay folding of the rRNA. Moreover, unexpectedly a tRNA is present in the E site of two assembly intermediates. Together, our study provides an architectural inventory of the distinct late assembly phase of the human 39S mitoribosome.

## Results

**Cryo-EM analysis of late mitochondrial ribosomal large subunit intermediates.** Both, MALSU1 and GTPBP10, a member of the Obg subfamily of GTPase and functionally conserved homolog of *E. coli* ObgE, are expected to act during the late phases of mtLSU assembly[23,24,30], and were thus chosen as baits for affinity purification. All particles were isolated using Flag-tag immuno-precipitation from human HEK293 cells and characterized by cryo-EM single-particle analysis. After processing, we obtained nine well-defined classes that represent five different folding states of the 16S rRNA with some additional variation regarding AF or tRNA association (Fig. 1, Supplementary Figs. 1–3). Not surprisingly, according to the local resolution estimation, they all displayed a highly ordered (high-resolution) solvent side, whereas a rather flexible intersubunit side was indicative of flexible regions waiting for the late maturation events to happen (Supplementary Fig. 4). However, all reconstructions displayed average resolutions between 3.1 Å and 5.7 Å allowing us to build de novo models or to perform rigid-body fitting of homology structures (Fig. 1, Supplementary Figs. 4 and 5, Supplementary Tables 1 and 2).

According to the folding states of the 16S rRNA, we classified them into five principle states (named 1–5) and arranged them in the most plausible order, most probably representing the sequence of the late maturation steps of the 39S ribosome. Two major states contained sub-states due to the different AF composition (named A to D) (Fig. 1 and Supplementary Table 2). State 1 already exhibited a 39S ribosome-like structure showing both the L1 and L7/L12 stalk, however, with an immature central protuberance. In addition, rRNA helices H64–65 and H67–71 of domain IV, and helices H80–H93 of domain V of the 16S rRNA were largely delocalized. State 2, which falls between State 1 and 3, displayed already close to mature positions of helices H80–88 of domain V of the 16S rRNA as well as rigid binding of bL33, leaving only helix H81 of the 16S rRNA unassigned (Fig. 1). From state 3 to 4, helices H80–H88 and H89–H93 of the domain V were folded in place sequentially, together with the central protuberance. In addition, the NSUN4–mTERF4 complex was observed interacting with the immature helices H68–71 of domain IV in state 4. Furthermore, state 4 showed bL36 in its mature position. The folding state of the 16S rRNA in state 3B, 3C, and 3D is the same as state 3A, yet, a major difference is the presence of a tRNA in the E-site (E-tRNA) and/or the MALSU1 complex. In detail, state 3A contained both the MALSU1 complex and the E-site tRNA, whereas state 3B and 3C contained either the MALSU1 complex or tRNA (Fig. 1 and Supplementary Fig. 3). Surprisingly state 3D contained neither one (Supplementary Fig. 3). Similar to state 3, state 5 also divides into two sub-states, state 5A and 5B with E-tRNA only present in state 5A. Notably, apart from the remaining association of E-tRNA and the MALSU1 complex, state 5A resembles already the mature 39S

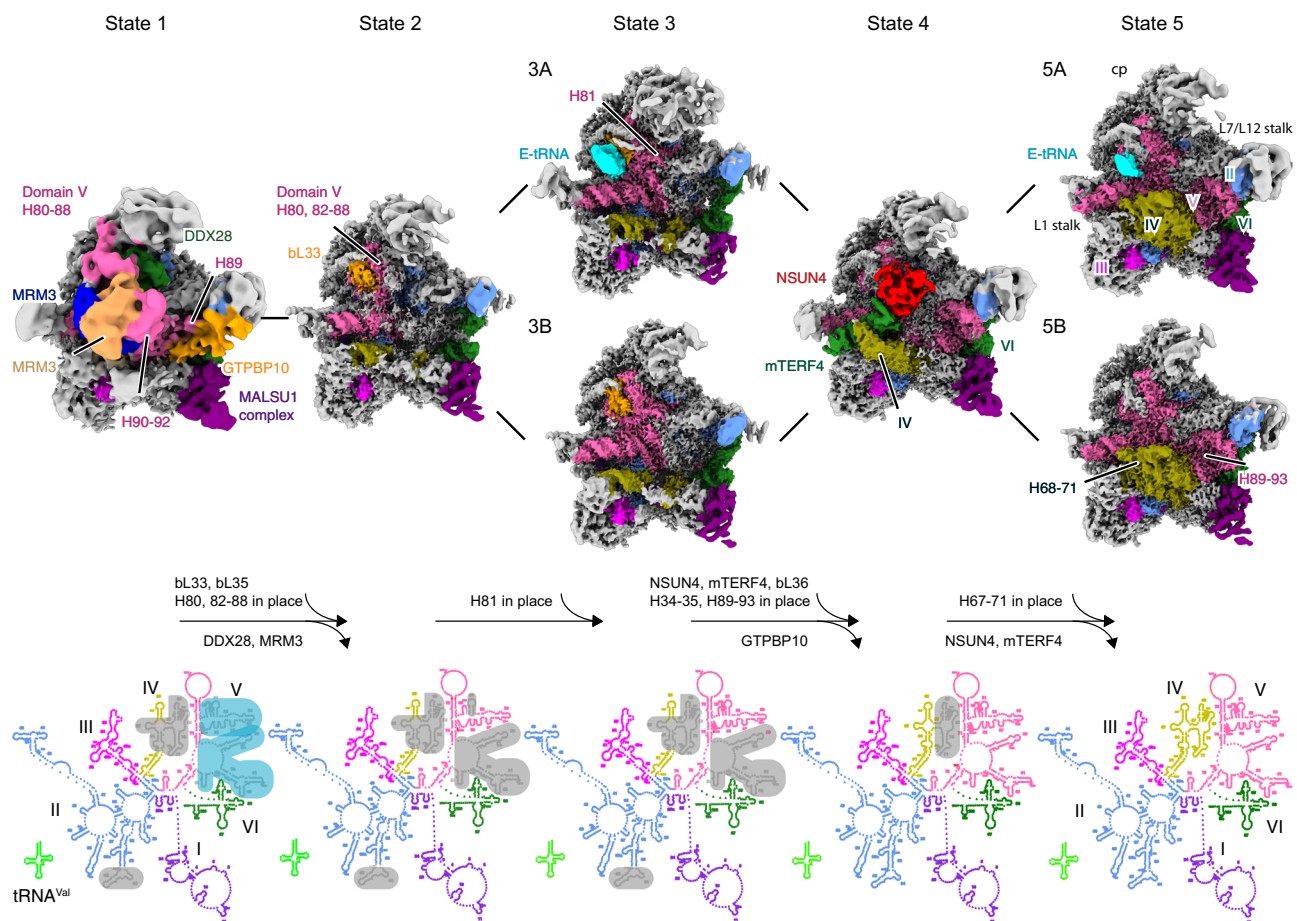

**Fig. 1 Ensemble of late human 39S mitoribosome assembly intermediates.** Cryo-EM maps of seven different intermediates of the 39S mitoribosome (top row) are shown with the 16S rRNA secondary structures corresponding to states 1–5 (bottom row). All assembly factors (MRM3: blue and sandy brown, DDX28: green, GTPBP10: orange, NSUN4: red, mTERF4: green, MALSU1 complex: purple) and six subdomains of the 16S rRNA are color coded (domain 1–6: purple, blue, magenta, yellow, hot pink, and green) and ribosomal landmarks indicated (cp, central protuberance). Immature or unstructured regions of the rRNA are shaded in light blue and gray in the secondary structure schemes, respectively. All maps were filtered to local resolution using Relion.

mitoribosome (Fig. 1). In conclusion, our structural analysis revealed several novel assembly intermediates representing the late maturation steps of the human 39S mitoribosome.

**DDX28 keeps the central protuberance immature**. Of the analyzed intermediates, State 1 showed the least mature conformation, characterized by the immature central protuberance and an immature PTC. However, at the same time it already showed the well-defined solvent side shell of the 39S mitoribosome (Fig. 1). Notably, a similar state has also been observed both, in vivo and in vitro for bacterial 50S pre-ribosomes[39–42]. In state 1, the core of the 39S mitoribosome, formed by domain I, II, III, and VI of the 16S rRNA, was well resolved and properly folded in its native mature conformation. Yet, besides rRNA helix H75, which forms the L1 stalk, almost the entire domain IV and domain V were still delocalized (Fig. 1).

The central protuberance consists mainly of helices H80–88 of domain V of the 16S rRNA and a valine tRNA. When comparing state 1 with the later state 5A, in which the 16S rRNA already adopted its mature conformation, the valine tRNA had already been recruited, whereas the helices H80–88 of domain V are completely distorted through interaction with the AF DDX28 (Fig. 2a). This AF is a DEAD box helicase that has been shown to localize to the RNA granules where the mitoribosome assembly occurs. DDX28 is known to interact with the 16S rRNA of the 39S mitoribosome and has been proposed previously to function during late assembly phases[26,27].

Although the local resolution is too limited as to reveal atomic detail, we can unambiguously rigid body fit a homology model of DDX28 into our state 1 based on the resolved secondary structure (Supplementary Fig. 5a). The assignment of this density to DDX28 is in agreement with mass spec analysis confirming its presence in the purified intermediates (Supplementary Data 1, 2). Notably, when inspecting the substrate binding pocket of DDX28 we could clearly trace an RNA density, corresponding to the region 1245–1251 of 16S rRNA domain V (Fig. 2b and Supplementary Fig. 5b). Thus, consistent with the published data, DDX28 is involved in late 39S mitoribosome assembly by exerting is helicase activity on the central protuberance[27].

Due to the binding of DDX28 to the central protuberance, helices H80–88 of domain V were kept in a substantially distant location from their mature position. Helix H86, when compared with the mature conformation, appeared simply shifted upwards, whereas rRNA helices H81, H82, and H88 were also rotated upwards, resulting in an open conformation of this entire subdomain (Fig. 2c). Two of the mitoribosomal proteins, bL33 and bL35, which interact tightly with the rRNA of this region in the mature conformation, were not yet recruited (Fig. 2d). Furthermore, the bound valine tRNA and the associating mitoribosomal proteins were observed in a rotated premature conformation (Fig. 2e). Taken together, the presence of DDX28 in state 1 results in the stabilization of the central protuberance in an immature conformation.

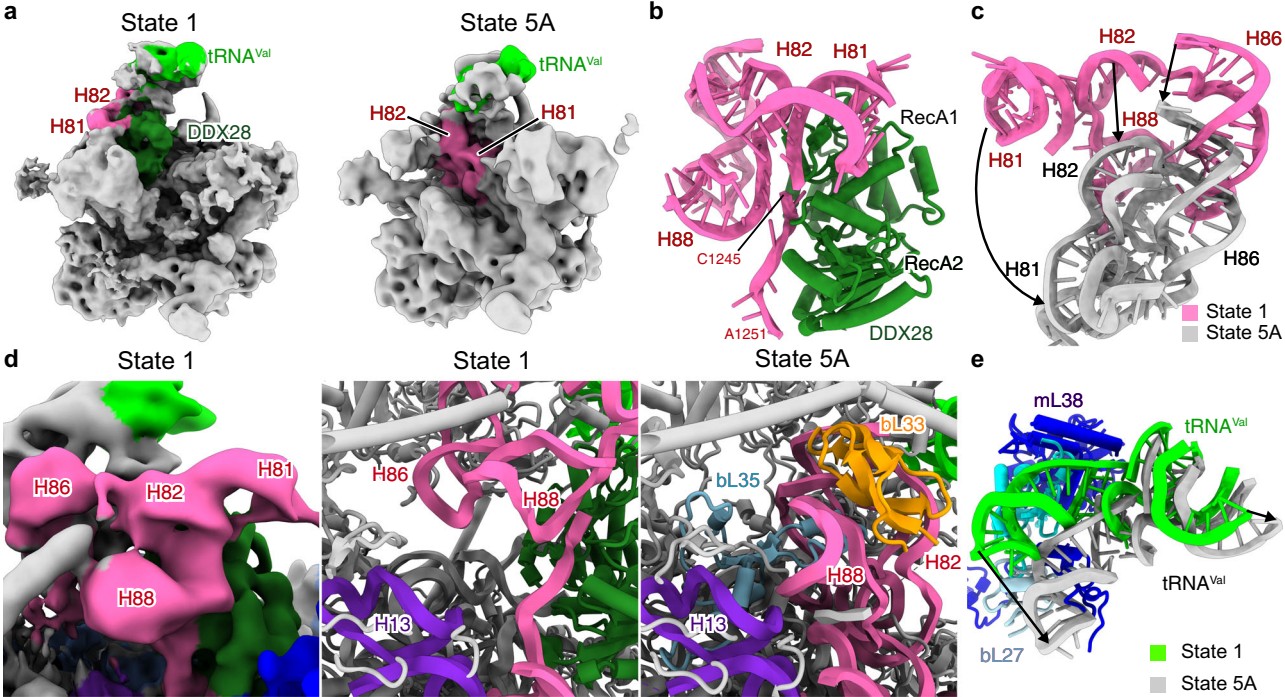

**Fig. 2 The immature central protuberance and DDX28 helicase in state 1. a** Overall positions of H80–88 of domain V of the 16S rRNA and DDX28. **b** DDX28 interacts with nucleotides 1245–1251 of domain V of the 16S rRNA. **c** Repositioning of H80–88 between state 1(pink) and 5A (gray). **d** Density showing void between H82 and H88 in state 1 (left), and models of state 1 and 5A showing incorporation of bL33 and bL35. **e** Comparison between state 5A and state 1, highlighting an immature conformation of the valine tRNA in the central protuberance of state 1.

**MRM3 and GTPBP10 keep the PTC immature**. Situated below the central protuberance and functionally most important in the large ribosomal subunit is the PTC region, which comprises the rRNA helices H90-93 of domain V. In state 1, however, this region is shifted outwards and located right in front of the immature helices H80–88 of domain V (Fig. 3a). This structural arrangement is stabilized by the methyltransferase MRM3 dimer (Fig. 3b). In addition, we observed a direct interaction between the N-terminal domain of one copy of MRM3 and the RecA2 domain of DDX28, which may indicate a coordination in the recruitment of these two factors (Fig. 3c and Supplementary Fig. 5c). MRM3 belongs to the SpoU methyltransferase family characterized by a classical C-terminal methyltransferase domain as well as a structured N-terminal domain. Two SpoU-like methyltransferases, MRM1 and MRM3, are present in human cells and function in mitoribosome assembly, but only MRM3 has been shown to modify G1370 in the A-loop in helix H92, which agrees perfectly with their assigned position (Fig. 3b)[15,17]. In contrast, MRM1 is known to modify G1145, a residue that could not be reached from the observed density[15]. Based on secondary structure, we can unambiguously fit MRM3 (Supplementary Fig. 5d), which positions the active center of methyltransferase domains right on top of the A-loop (Fig. 3b). In agreement with this assignment, a homologous methyltransferase was observed also in the mitochondrial assembly intermediate from *Trypanosoma brucei* in an essentially identical conformation (Fig. 3b). However, the location of rRNA domain V is dramatically different, which highlights the specificity of the human mitoribosome assembly pathway (Supplementary Fig. 6)[36]. Although at the given resolution we can only tentatively assign a MRM3 homodimer to the observed symmetric density, it would be reminiscent to many members of the SpoU family methyltransferases that form functional homodimers[43] and is thus the most plausible interpretation. In any case, we find that the mode

of modification of the A-loop is conserved between species in mitochondria. In eukaryotes, like in mitochondria, it is executed by a methyltransferase protein (Spb1 in yeast) during late assembly and not as many modifications by a snoRNP in an earlier phase[44]. Apparently, in bacteria a relative of MRM3, FtsJ, executes this modification, probably in a similar way[45].

Also in state 1, helix H89 of domain V was still located in the vicinity of its mature position, yet shifted outwards and stabilized by GTPBP10 (Fig. 3d, e). Consistent with our structural analysis, biochemical studies have shown previously that this AF directly interacts with 16S rRNA between states containing DDX28 or NSUN4/mTERF4, respectively[23,24]. We observed the GTPBP10 bound to the sarcin-ricin-loop (SRL) and the stalk base, which is in agreement with the findings for ObgE, the bacterial homolog of GTPBP10 (Fig. 3f)[46]. Notably, also in eukaryotes the ObgE-type GTPase Nog1 binds to and displaces the rRNA helix H89 as observed in yeast[47]. Thus, the GTPBP10/ObgE regulation may be conserved and, like the translational GTPases, activation of GTP hydrolysis of GTPBP10 might involve the SRL of the mtLSU. However, ObgE, which was used as a bait to purify native *E. coli* intermediates, was observed to associate only with later states of assembly and thus has a different function in 50S subunit assembly compared to GTPBP10 in 39S assembly[42].

**Premature association of E-site tRNA in states 3 and 5**. During the late maturation, we observed four different sub-states of the state 3, state 3A–D. They all show the same immature conformation of the 16S rRNA as the previously reported 39S mitoribosome intermediate[31]. However, only states 3A and 3B, but not 3C and 3D showed the MALSU1 complex. Surprisingly, we also found premature incorporation of a tRNA into the E-site in states 3A and 3C (Fig. 4). Due to steric hindrance, binding of the NSUN4–mTERF4 complex in state 4 prevents binding of the E-site tRNA in this state (Fig. 1). Apparently, it can rebind in

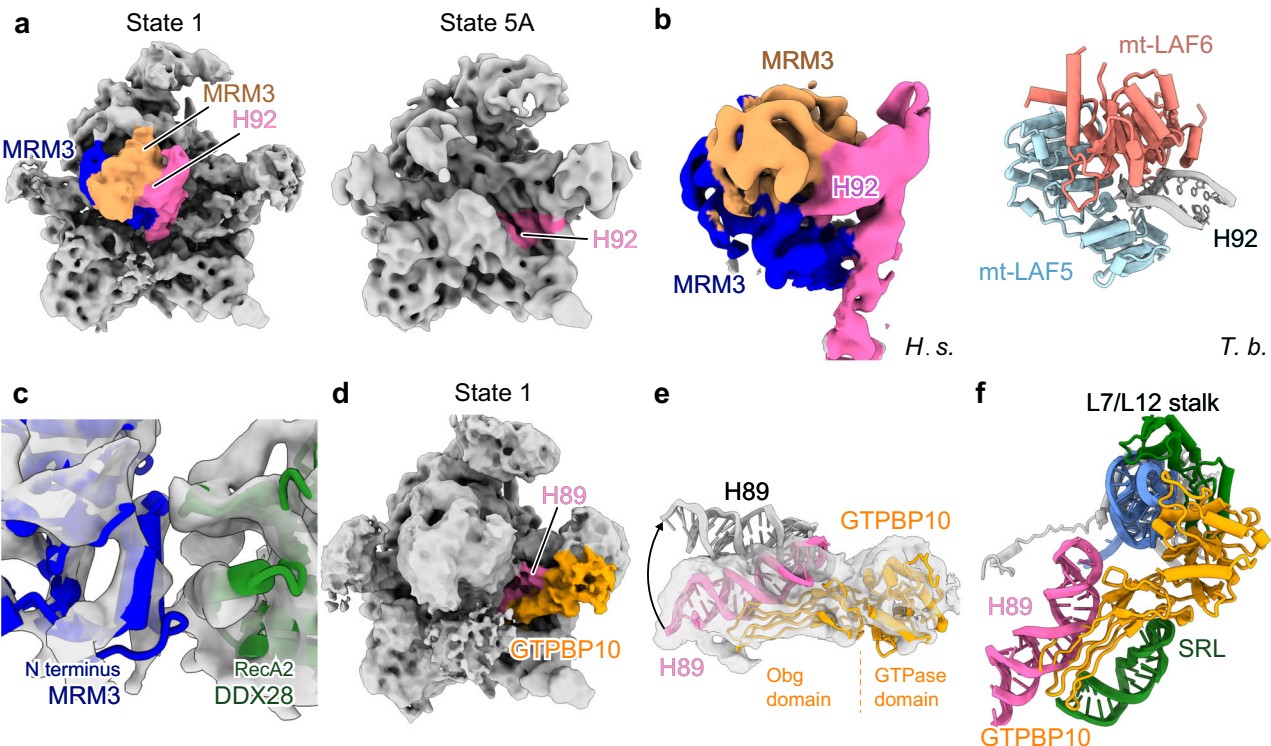

**Fig. 3 State 1 displays an immature PTC with GTPBB10 and MRM3 complex bound. a** Overall position of domain V in state 1 (left) and 5A (right). H92 of the 16S rRNA and MRM3 dimer (in blue and sandy brown) are labeled. **b** Comparison between the human MRM3 homodimer bound to H90–H92 and the mt-LAF5-mt-LAF6 heterodimer (in light blue and salmon, respectively) bound to H92 in maturing *Trypanosoma brucei* (PDB: 6YXX) large mitoribosomal subunit. **c** Interaction of N-terminal domain of MRM3 with the RecA2 domain of DDX28. **d** Overview of GTPBP10 (orange) and H89 (hot pink) in state 1. **e** H89 adopts an immature conformation in the presence of GTPBP10 (pink, H89 in state 1: gray, mature H89). **f** GTPBP10 interacting with H89 and the SRL of the 16S rRNA and the L7/L12 stalk. *H. s.*: *Homo sapiens*, *T. b.*: *Trypanosoma brucei*.

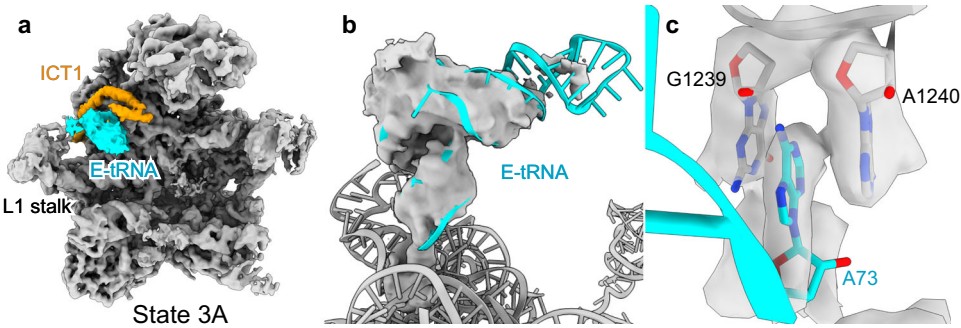

**Fig. 4 Binding of E-site tRNA to immature mitoribosomes. a** State 3A shows an E-site tRNA (cyan) interacting with ICT1 (orange). **b** Cryo-EM density of the E-site tRNA superimposed on the model indicating flexibility of the anticodon stem. **c** Detailed view of the CCA end of the deacylated tRNA with the terminal A stacking between 16S rRNA bases.

state 5A or, alternatively, a direct transition from state 3 to 5 is possible. In any case, this finding contradicts the commonly accepted principle that, in general, premature binding of translation factors or components is prevented during ribosome assembly. The observed binding of E-site tRNA could be accidental and without any functional relevance. Yet, another possibility is that mitochondria take advantage of the translational components already present in the mitochondrial matrix to either stabilize assembly intermediates or to monitor the folding state of the central protuberance even before its full maturation.

**The NSUN4–mTERF4 complex grips the immature domain IV.** In the observed state 4, the domain V of the 16S rRNA

appeared already almost fully matured, leaving only helices H67–71 of the domain IV immature (Fig. 1). In addition, we observed an additional density in this state and, based on secondary structure information, we could unambiguously fit the NSUN4–mTERF4 X-ray structure (Fig. 5a and Supplementary Fig. 5e). Recent structures have shown that NSUN4 forms a stable complex with mTERF4, suggesting that mTERF4 targets NSUN4 to the rRNA and regulates its activity[33,34]. Although NSUN4 only modifies residue C911 in the mtSSU, it was also shown to interact with the mtLSU and, thus, has a dual function in both mtSSU and mtLSU assembly[16].

Consistent with these published results, we observed the NSUN4–mTERF4 complex directly above the peptide transfer

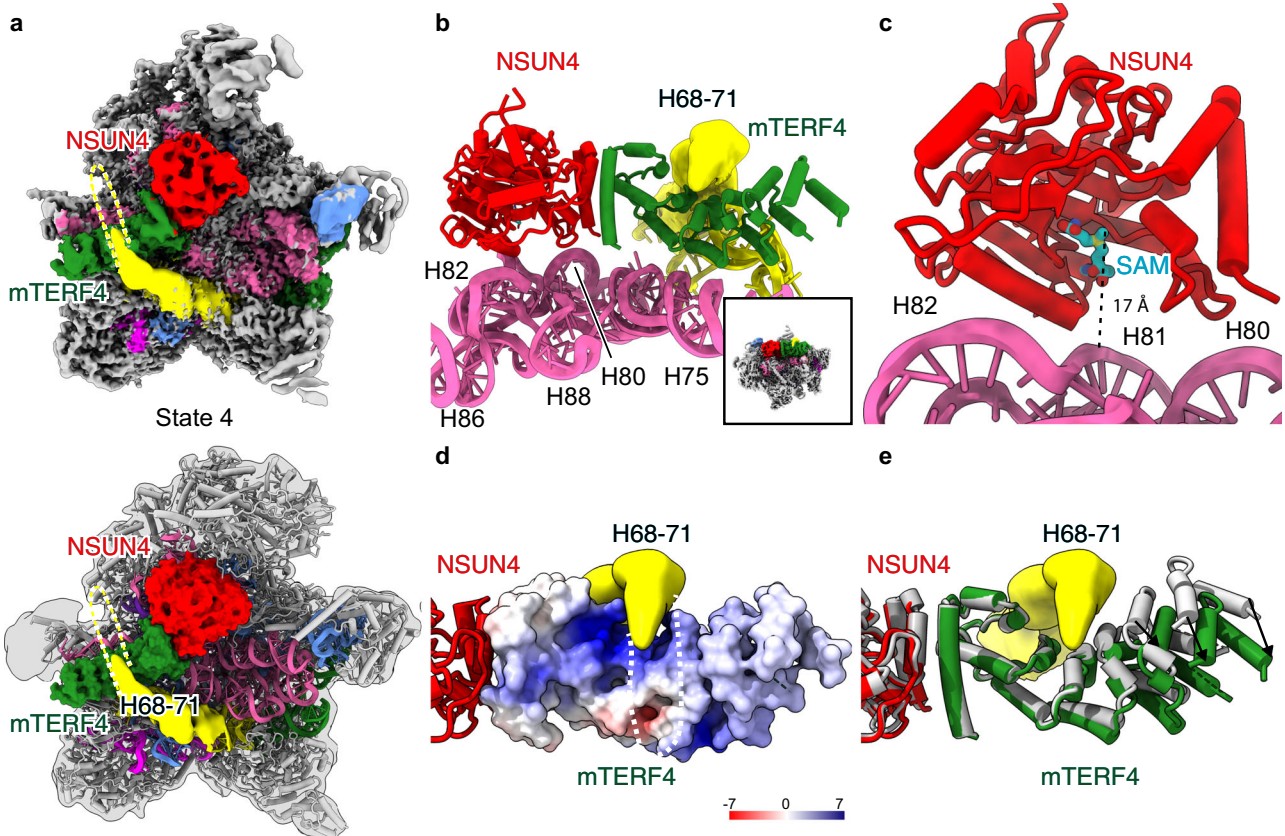

**Fig. 5 Immature rRNA helices H68–71 and NSUN4–mTERF4 complex in state 4. a** View onto the intersubunit side of the map (top) and model (bottom) of state 4. The NSUN4–mTERF4 complex is shown as a surface model. **b, c** close-up views of the interaction between the NSUN4–mTERF4 complex and the 16S rRNA, illustrating the distance between the catalytic center of NSUN4 and the 16S rRNA. To locate the active center of NSUN4 S-adenosyl methionine (SAM) was docked from the crystal structure (pdb:4FZV). **d** rRNA helices H68–71 interact with mTERF4 which is shown as surface and colored according to its surface potential. **e** Comparison of unbound mTERF4 (PDB: 4FZV, gray) with mTERF4 bound to the mitoribosome indicating conformational changes. Helices H68–71 in (**a, d**) are shown as yellow density with dashed lines.

center (PTC), with the NSUN4 subunit very close to the helix H80–88 region of domain V and the mTERF4 subunit sandwiched between helices H68–71 and H75 of domain IV (Fig. 5a, b). The active site pocket of NSUN4 was located in juxtaposition to helix H82 of the 16S rRNA, however, with a too large distance as to reach it for modification (Fig. 5c). This is in agreement with the finding that NSUN4 methylate 12S but not 16S rRNA[16]. Notably, in this conformation NSUN4 would also be prevented from accessing the 12S rRNA of the 28S mitoribosome, thereby excluding modification activity as a potential assembly check-point upon subunit joining. Thus, the NSUN4–mTERF4 complex may rather have a scaffolding function on the mtLSU and likely serves to check and stabilize the maturation state of the domain V of the 16S rRNA while keeping domain IV in a largely immature position. Specifically, the mTERF4 subunit maintains helices H68–71 of domain IV in an immature conformation. This RNA segment is the last building block required to complete the PTC. As suggested before, it is likely that a positively charged patch on the surface of mTERF4 is used to interact with this region (Fig. 5d)[34]. When comparing mTERF4 in our complex with the RNA-free X-ray structure, we observed that the N-terminus of mTERF4 undergoes a conformational change towards a more opened conformation (Fig. 5e). Thereby, the domain IV rRNA is held in the concave middle of mTERF4, thus keeping this region far away from its mature location.

## Discussion

Based on the conformation and maturation states of the 16S rRNA, our ensemble of intermediate structures purified with the assembly factors MALSU1 and GTPBP10 can be put in a sequential order representing the late maturation path of the human 39S mitoribosome (Fig. 6). In the early intermediate state 1, the terminal domains of 16S rRNA, domains I, II, III, and VI first assemble to form the solvent side core of the 39S mitoribosome, only leaving the middle domains IV and V immature. This is achieved with the employment of assembly factors DDX28, MRM3, and GTPBP10, which each bind and stabilize different parts of these two rRNA domains in immature conformations, respectively. Upon further progress, DDX28 together with the MRM3 complex leaves the intermediate with the helices H80–88 of domain V matured. Thus, the further matured conformation of the central protuberance may be checked and stabilized by binding of the E site tRNA. In this state (state 3), although we could not rule out that GTPBP10 might be dissociated during preparation, GTPBP10 is expected to still be associated with the flexible and therefore invisible 16S rRNA helix H89 since the same state was obtained using different approaches. Thus, GTPBP10 most likely moves together with H89 without providing visible density, but it is still associated with the particles. Subsequently, upon dissociation of GTPBP10, which keeps helices H89–93 of domain V immature, allows for the maturation of domain V to conclude, leaving only helices H68–71 of domain

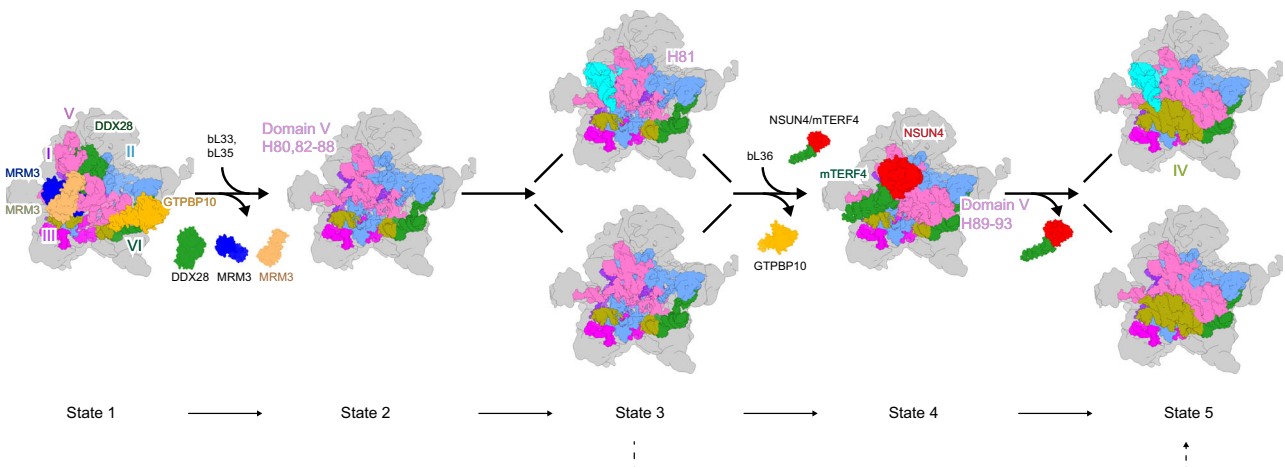

**Fig. 6 Late assembly steps of the 39S mitoribosome.** Cartoon depicting the late assembly transitions during the 39S mitoribosome maturation. A possible bypass from state 3 directly to state 5 is indicated with dash line. Assembly factors and the six subdomains (I–VI) of the 16S rRNA are color coded. For detailed description see text.

IV in an immature conformation. This intermediate state is stabilized by the interaction with the NSUN4–mTERF4 complex, which at the same time, can also serve as a final check-point to monitor the folding of the domain V. At this point, only dissociation of the NSUN4–mTERF4 complex is required to allow for the final rearrangement of rRNA domain IV and thus formation of the mature 39S mitoribosome. The observed ensemble of intermediates provides little evidence for alternative pathways when focusing on the conformational transitions of the rRNA, however, we cannot exclude that they exist, possibly under stress conditions. Only the transition from state 3 to 5 may happen directly by bypassing state 4, which would imply that maturation of remaining immature rRNA domains IV and V can occur concomitantly with the MTERF4–NSUN4 complex being dispensable.

This ends-to-middle order of rRNA domain accommodation is conserved in 23S/25S rRNA assembly of the cytoplasmic 50S/60S subunit maturation, in which domains IV and V are also kept immature until the late stages, with H69 of the 25S rRNA being the last to be completed[32,48,49]. A similar non-canonical sequence has also been observed with 18S rRNA assembly of the cytoplasmic 40S small subunit maturation[50]. Taking this into account, we conclude that an initial assembly sequence of ribosomal RNA assembly, which is seeded by the 5′ and 3′ ends, presents a universal mechanism, conserved from prokaryotes to eukaryotes and within eukaryotes shared between cytoplasmic and organellar (mitochondrial) ribosomes (Supplementary Table 3). We speculate that one reason for this to occur is the implementation of an efficient and reliable way to ensure the completeness of the rRNA transcription, thereby mitigating unnecessary energy expenditure on incomplete rRNA intermediates. The same principle may also apply to the 12S rRNA assembly of the 28S small mitoribosomal subunit.

Through structural analysis, we identified DDX28, MRM3, GTPBP10, NSUN4, and mTERF4 as late assembly factors on the 39S mitoribosomal intermediates. Although we do not have sufficient resolution to confirm all the assignments based on side chain information, we are confident in our assignments for the following reasons: First, the presence of all factors in the isolated intermediates was confirmed by mass spectrometry (MS) (Supplementary Data 1, 2). Second, our maps with density corresponding to DDX28, MRM3, and the NSUN4–mTERF4 regions have sufficient local resolution to assign the secondary structure.

Thus, we can fit either homology models or crystal structures of the respective factors. NSUN4 and mTERF4, for example, form a heterodimer and here the X-ray structure of the entire complex fits the density after small adjustments (Supplementary Fig. 5). Third, the DDX28 density clearly adopts a DEAD box helicase fold and DDX28 is the only DEAD box helicase known to be involved and present in the MS analysis. Fourth, it is known that MRM3 modifies G1370 on the A loop which is in perfect agreement with the position of the MRM3 dimer density after docking[15,17]. Last, the GTPBP10 density has an ObgE type GTPase shape and interacts with the L7/L12 stalk and SRL, which is commonly observed for this protein family both, in prokaryotes and eukaryotes. MTG2 as a second mitochondrial ObgE-like GTPase also participates in 39S mitoribosome assembly, however, it is not present in our sample according to MS analysis (Supplementary Data 1, 2). We are thus confident that our density assignments are highly likely to be correct.

Regarding the biogenesis of the eukaryotic 80S ribosome, it is a well-accepted concept that during the assembly all functional regions are kept immature or masked in order to prevent premature association of translation factors, such as mRNA or tRNA. To our surprise, however, this appears not to be valid for the mitoribosome assembly since we observed the premature incorporation of mature deacylated tRNA in the E site in two of the intermediates, which only been observed on mature 39S mitoribosome so far[51]. One possibility is that such early binding of tRNA serves a test drive for functionality of the E-site, similar to the suggested test drive of the pre-40S by engaging the mature 60S subunit to from a first complete 80S-like ribosome in yeast[52]. However, when taking also the replacement of 5S rRNA by the valine tRNA in the central protuberance into account, it appears that mitoribosome architecture and assembly has evolved to maximize the usage of factors that are already present in mitochondria. This limits the requirements for genes encoded in the mitochondrial DNA (e.g., for the 5S rRNA) and also for the import of nuclear-encoded factors. In agreement with this idea, we also observe the repurposing of the NSUN4–mTERF4 methyltransferase complex. This enzyme is primarily active in the modification of the small subunit rRNA[16]. But instead of displaying the function as a methyltransferase as for the 12S rRNA, on the 16S rRNA it may only sense the maturation state of the large subunit and triggers the final maturation step of domain IV rRNA. Interestingly, we observed before that RNA modifying

enzymes can be repurposed in order to act on another ribosome biogenesis substrate without displaying its original enzymatic activity. Nop1, for example, exerts methyltransferase activity, yet, in the yeast 90S pre-ribosome, Nop1 also cannot access to the 18S rRNA, thus can only serve as a scaffold protein to maintain the U3 snoRNP structure for coordinating the pre-40S ribosome[53]. In conclusion, a reoccurring theme during ribosome biogenesis is the evolution of modifying enzymes to adopt additional roles beyond their enzymatic activity, mostly gaining functions as rRNA chaperones or scaffold proteins.

Recently, several kinetoplastid mitoribosomal assembly intermediates were reported, which allow for the comparison of mitoribosome assembly pathways between species (Supplementary Table 4)[36–38]. Despite the presence of some highly-conserved factors, the kinetoplastid uses numerous species-specific assembly factors which are the main contributors to shape the pre-ribosomal intermediates. As a result, the immature arrangement of 16S rRNA in our state 1 is completely different compared to the kinetoplastid intermediates, indicating substantial differences in the mitoribosome assembly pathways between species (Supplementary Fig. 6). This is in line with the observation that due to distinct evolutionary factors and in contrast to the cytoplasmic 80S ribosome, mitochondrial ribosomes display dramatic differences between species. Yeast and plant mitoribosomes can even be larger than cytoplasmic 80S ribosomes, whereas some insect and vertebrate mitoribosomes, such as the human one, are extremely small and retained only their rRNA core. Our study thus provides information on the mitochondrial ribosome assembly pathway which is specific for human cells and can be directly related to the context of human mitochondrial disease.

## Methods

**Molecular cloning and generation of cell lines**. Human *MALSU1* and *GTPBP10* genes were amplified from a human cDNA library and inserted into a modified pcDNA5/FRT/TO vector which has a C-terminal Strep-Flag tag. The commercial HEK Flp-In 293 T-Rex cell line was used to generate stable cell lines. In detail, the cells were split one day before the transfection, and transfected with the MALSU1 or GTPBP10 plasmid and a helper plasmid pOG44 using PEI according to the manufacturer's guidance. Two days after transfection, cells were split and transfered into a new plate with normal DMEM medium supplied with 10% FBS, 200 µg/ml hygromycin B, 10 µg/ml blasticidin and 1× penicillin/streptomycin. After several days of selection, the remaining cells were tested for protein expression and frozen in liquid nitrogen.

**Sample purification**. The MALSU1 or GTPBP10 cells were cultured in normal DMEM medium with 10% FBS. To induce the expression of the tagged protein, one day after splitting, a final concentration 1 µg/ml of tetracycline was added to induce expression for 24 h before harvest. A total of 25 dishes (15 cm) of cells were scratched down and washed once with cold PBS buffer. Lysis buffer (50 mM HEPES pH 7.4, 150 mM KCl, 5 mM MgCl₂, 1 mM DTT, 0.5 mM NaF, 1 mM Na₃V₃O₄, 1× protease inhibitor mix) was added to resuspend the cell. After lysis with 10 strokes using a douncer, the cell lysate was kept on ice for 10 min. Subsequently, the cell lysate was cleared by centrifugation at $10,000 \times g$ at 4 °C for 25 min and immediately incubated with pre-equilibrated anti-Flag affinity beads for 2 h. After incubation, the beads were transferred into a new small column and first washed once with one column volume lysis buffer and then three times with wash buffer (20 mM HEPES pH 7.4, 150 mM KOAc, 5 mM Mg(OAc)₂, 1 mM DTT). After washing, elution was performed by three times incubation with 200 µl elution buffer (0.2 mg/ml 3×Flag peptide in washing buffer) for 15 min. The final elution was concentrated using 100 kDa cut-off Amicon concentrator, and the final concentration was measured using a NanoDrop photometer.

**Electron microscopy and image processing**. Before freezing, a final concentration of 0.05% Nikkol was added to improve the ice quality. 3.5 µl of the sample (MALSU1 sample: OD₂₆₀ = 0.8, GTPBP10 sample: OD₂₆₀ = 1.5) was directly applied onto 2 nm carbon pre-coated R3/3 holey copper grids (Quantifoil), blotted for 3 s and plunge-frozen in liquid ethane using a Vitrobot Mark IV. Cryo-EM data was acquired on a Titan Krios transmission electron microscope (Thermo Fisher Scientific) operated at 300 kV under low-dose conditions (28 e⁻ Å⁻² in total) with a nominal pixel size of 1.084 Å on the object scale using EPU 2 software. A total 12,430 micrographs (for MALSU1 sample) and 9834 micrographs (for GTPBP10 sample) were collected on a Falcon II direct electron detector. The

original frames were aligned, summed and drift-corrected using MotionCor2[54]. Contrast transfer function parameters and resolution were estimated for each micrograph using CTFFIND4[55] and Gctf[56], respectively. Micrographs with an estimated resolution below 4 Å and astigmatism below 5% were manually screened for contamination or carbon rupture. As a result, a total 11,388 micrographs (for MALSU1 sample) and 9452 micrographs (for GTPBP10 sample) were selected. Particle picking was carried out using Gautomatch with a low pass filtered mature human 39S mitoribosome as a reference[4]. After 2D classification, a total 785,354 particles (for MALSU1 sample) and 1,813,074 particles (for GTPBBP10 sample) were submitted to 3D refinement, 3D classification, CTF refinement as shown in Supplementary Fig. 1 using Relion 3.1[57]. During the revision of this paper, we collected another dataset using uL17 as bait. This dataset will be discussed somewhere else. However, a small portion of the dataset also contains the desired state 1. Together with data of state 1 from the other two datasets, we combined all of them together, and did another round of 3D classification focusing on MRM3/DDX28 region. The resulting class which shown the best resolved MRM3/DDX28 region was picked to get the final reconstruction of state 1 (Supplementary Fig. 2). All the final map was filtered according to the local resolution in Relion.

**Model building and refinement**. In general, the models of human mature 39S mitoribosome (PDB:3J9M)[4] and two intermediates (PDB:5OOL, 5OOM)[31] were used for initial rigid body docking into the density. Since the 39S mitoribosome part of all states resembled very well the mature 39S mitoribosome, the mature human 39S mitoribosome model could be fitted as rigid body. The immature region in rRNA domain IV and V were either simply removed or manually adjusted in Coot[58] to better fit the density. The MALSU1 complex was taken from PDB:5OOL and rigid body fitted into every state[31]. For state 1, the overall resolution was not sufficient to build a molecular model. DDX28 was fitted as a rigid body using a homology model generated using the SWISS-MODEL server based on PDB: 4W7S[59,60]. After fitting, small adjustments were done manually in Coot. MRM3 and GTPBP10 were also rigid body docked as homology model generated using the SWISS-MODEL server based on PDB: 4X3M and 1LNZ[61], respectively. However, due to the low local resolution, no adjustments could be further done, and, in the final model of state 1, only poly-Ala models are provided. In the states 3A and 5A, the E-site tRNA did not have sufficient resolution to build a molecular model, thus a previously observed mitochondrial tRNA was used to rigid body fit into the density[9]. The position A73 was manually built in Coot. For state 4, the crystal structure of NSUN4–mTERF4 complex was used to dock into the map, with a small manual adjustment of mTERF4[33,34]. Also here, due to low local resolution of these two factors, only poly-alanine models were provided. Model refinement was carried out using Phenix.real_space_refine, and final models were validated using MolProbity[62,63] inside of the Phenix software suit. Maps and models were visualized and figures created with ChimeraX[64].

**Reporting summary**. Further information on research design is available in the Nature Research Reporting Summary linked to this article.

## Data availability

The EM density maps have been deposited in the Electron Microscopy Data Bank under accession codes EMD-12919 (state 1), EMD-12920 (state 2), EMD-12921 (state 3A), EMD-12922 (state 3B), EMD-12923 (state 3C), EMD-12924 (state 3D), EMD-12925 (state 4), EMD-12926 (state 5A), EMD-12927 (state 5B), and the coordinates of the EM-based models have been deposited in the Protein Data Bank under accession codes PDB 7OI6 (state 1), PDB 7OI7 (state 2), PDB 7OI8 (state 3A), PDB 7OI9 (state 3B), PDB 7OIA (state 3C), PDB 7OIB (state 3D), PDB 7OIC (state 4), PDB 7OID (state 5A), PDB 7OIE (state 5B). Other coordinates of the used models in this study are available under accession codes: PDB 3J9M, PDB 5OOL, PDB 5OOM, PDB 4W7S, PDB 4X3M, and PDB 1LNZ. A Life Sciences Reporting Summary for this paper is available. Supplementary datasets 1 and 2 are provided with this paper.

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

## Acknowledgements
The authors thank S. Rieder, C. Ungewickell, and A. Gilmozzi for technical assistance, L. Kater for discussions and critical comments on the manuscript. This research was supported by grants from the Deutsche Forschungsgemeinschaft (GRK1721) and by an European Research Council (ERC) Advanced Grant (HumanRibogenesis) to R.B.

## Author contributions
J.C. and R.B. designed the study. J.C. performed all the experiments and O.B. collected cryo-EM data. J.C. and R.B. analyzed the structures, interpreted results, and wrote the manuscript.

## Funding

## Competing interests
The authors declare no competing interests.
