## [Peer Review File · Nature Communications]

REVIEWER COMMENTS

Reviewer #2 (Remarks to the Author):

This manuscript from Roland Beckmann's group describes structures of late intermediates in assembly of the human mitochondrial large ribosomal subunit. The authors affinity-purified these particles using epitope-tagged late assembly factors, then determined the structures of these assembly intermediates at 3.1 to 4.3 Angstroms resolution by cryo-electron microscopy. They identified four principle intermediates A-D, plus five more "substates". Interestingly, they observe a phenomenon similar to what was previously observed for bacterial and yeast large subunit assembly, that the 5' and 3' domains of rRNA comprising the solvent-exposed side of the subunit are stably assembled first. Then rRNA domains IV and V complete their maturation to form the subunit interface, where the functional centers (e.g. the PTC) are located. Thus, this mode of pre-rRNA maturation seems to be conserved. The manuscript also describes the location, structure, and potential functions of six assembly factors. The DEAD box protein DDX28 is bound to the immature CP and appears to prevent maturation of helices H80-H88. The methyltransferase MRM3 and GTPase GTPBP10 bind to helices H90-H93 and appear to delay maturation of this portion of the peptidyltransferase center. Surprisingly two intermediates contain a tRNA bound to the E site, perhaps performing a functional "test-drive". Finally, the NSUN4-TERF4 complex maintains helices H68-H71 of domain IV in an immature conformation, perhaps functioning as a final checkpoint.

I have no technical concerns about the manuscript. The results are very timely and of broad interest. However, I would like to offer several suggestions to make the manuscript more clear, and with all due respect, perhaps more interesting.

(1)Abstract: The phrase "Besides the MALSU1 complex, we identified..." seems somewhat abrupt. Readers may not yet know that the MALSU1 complex was used as a bait for purification. Thus, it seems strange to begin the sentence with this phrase. Perhaps, write instead: " Besides the MALSU1 complex used as bait for affinity purification, we identified..."

(2)Introduction: The last paragraph of the Introduction seems a bit too dry. Rather, I suggest that the authors highlight here their most interesting discoveries, e.g. how conserved the rRNA maturation pathway is, that several assembly factors appear to function to delay folding of the rRNA, and that there is a tRNA present in the E site.

(3)Introduction, lines 48-50: Is the mitochondrial ribosome assembly pathway "more complex," or is it simply different from other assembly pathways? or both? If more complex, in what sense?

(4)Introduction, lines 54-58: This list of some of the assembly factors reads like a laundry list...why are these listed and not others? And, will nonexpert readers appreciate these specific names? Perhaps, it would be more clear to simply state the different functional classes of assembly factors that have been found, namely, RNA helicases, methyltransferases, and GTPases. This lengthy list also distracted me from initially focusing on the six assembly factors discussed in the Results.

(5)Introduction, lines 90-91, and Figure 1: Can the authors make more clear why they segregated the substate structures from the four major classes of particles? For example, are the latter more abundant? More different? Having two separate sets was initially distracting, to forecast the pathway.

(6)Figure 1: As mentioned above, might the authors reorganize the structures into the order that they believe they mature? And, it might be useful to emphasize even more, somewhere in the manuscript (Discussion?), whether or not there is evidence for alternative pathways as seen in bacteria.

(7)Figure 1: the bright yellow domain IV rRNA structure is attractive, but difficult to see, especially the labels for helices...perhaps darken the yellow to mustard color?

(8)Figure1: Immature or unstructured rRNA is shown in gray. This is an important distinction...perhaps use two different colors (or two shades of gray)?

(9)Figure 2: Likewise, the pink is attractive but the labels are tricky to read...darken? (The red bL35 is difficult to see...perhaps light blue?

(10)It is potentially interesting that the DEAD box protein DDX28 appears to physically block maturation of the CP. Is this one of the first examples for such a (potential) function for a DEAD box protein, i.e., different from the RNA or RNP remodeling function ascribed to helicase activities? Seems interesting to me. On the other hand, the authors state on line 276 that "DDX28 remodels the intermediate ..." . Has it been proven (in the referenced work) that DDX28 in fact remodels the pre-ribosome in vivo, via an RNA helicase activity? If so, this needs to be made a little more clear. Instead, or in addition, might ATP hydrolysis propel DDX28 off of the pre-ribosome to enable subsequent movement of the CP?

(11)Lines 145-147: the word "distorted" conveys a negative (mutant?) tone to me, i.e., not a normal part of the wildtype folding pathway...maybe just use the word "immature"?

(12)Figure 3 and lines 179-180: The propinquity of the N terminal domain of MRM3 and the RecA2 domain of DDX28 suggests that MRM3 might regulate or recruit DDX28. What might further support this idea?

(13)Line 202: Typo: Sbp1 should be Spb1. I think there may also be a bacterial relative of MRM3 that modifies the A loop: FtsJ? JBC vol 275 p. 16414 (2000).

(14)Lines 205-218, about the GTPase GTPBP10: The authors might add that, in addition to the bacterial GTPase ObgE, the yeast GTPase Nog1 also binds to and displaces helix 89.

(15)Starting with line 215: It is fascinating that a tRNA is found in the E site in two apparently nonconsecutive assembly intermediates B and D. The authors do suggest that instead of maturing through state C, state B might directly transition to state D, or that either might occur. Perhaps the authors might elaborate further in the Discussion about alternative assembly pathways. In addition, the authors discuss, correctly I believe, that it is unusual to see interaction of a translation factor with a potential active site in an immature ribosome. Might the authors relate this to others' work describing test driving of functional centers?

(16)Lines 264-286: this last section of the Results nicely summarizes a significant fraction of the authors' findings in this manuscript. Therefore, might it better fit at the beginning of the Discussion? (Easy to change!)

(17)Line 273: The term "reversed order" referring to the observation that the terminal domains of rRNA mature before the middle domains seems a little confusing to me. Could the authors use a better description?

(18)The last section of Results nicely summarizes what was found about each late assembly factor, but omits mention of MRM3 and factor X.

(19)Discussion, lines 303-306. The authors reference prior work showing that the order of rRNA accommodation that they observe in mitochondrial large subunit assembly, maturation of domains containing the two ends of the RNA before the middle portion, also occurs in bacteria and yeast. The authors should also reference another publication that demonstrated this order of maturation of domains in the large subunit: (Gamalinda et al., Genes and Development 28,198-210 (2014)).

Sincerely,
John Woolford

Reviewer #3 (Remarks to the Author):

In the present paper Chen and colleagues analyzed the assembly of the human 39S subunit of the mitoribosome. Assembly intermediates were purified from cells via a Flag tag introduced at assembly factors MALSU1 and GTPBP10, respectively, and subjected to cryo-EM. From both specimen combined the authors obtained 9 cryo-EM maps of assembly intermediates of the 39S subunit at 3.1-4.3 Å resolution informing on late assembly steps. The authors can follow transitions of the 16S rRNA involved in maturation of the central protuberance and the peptidyltransferase center. They can also identify several assembly factors although insights here are hampered by low local resolution. The results are interesting and provide new insights into the important assembly pathway of the human mitoribosomes. However, there are a couple of points the authors may want to consider to improve their manuscript.

Specific Points:

1. The results part is difficult to follow and appears cluttered. In the abstract four distinct late state intermediates are mentioned, according to the beginning of the results section "nine well defined classes which showed clearly different folding states of the 16S rRNA" were obtained and according to supp. Fig. 1 there are even more reconstructions that are not discussed (states D1, D2) . It may help, to explicitly state which states were derived from which specimen and to provide an overview about the assembly factor composition of the various states, maybe in form of a table.
2. The figure legends should be update to explain the labels. In Fig. 1 for example, the lines connecting the states are not explained, in Fig. 5 a legend for the coloring of surface potential is missing.
3. It may be also beneficial to define upfront what a state is and what a substate. States A1 and B1 contain more factors, but A2 and B3 appear to be more abundant and can thus be considered the dominating substates.
4. From a technical point of view it is unclear, if the authors have really fully exploited the potential of method. In general the density corresponding to the assembly factors appears to be poor limiting insights into the detailed molecular interactions. The only state that appears to have density for GTPBP10 is state A1. Even in the GTPBP10 affinity purified specimen, where most of the complexes are expected to have GTPBP10, A1 is present to about 1% only (15% of 7.4%). Thus 99% of the complexes have lost the factor during preparation. It may be worthwhile to optimize the purification protocol and to explore crosslinking to improve complex stability. Also the application of specialized image processing methods such as 3D variability analysis implemented in the cryoSPARC software package may be beneficial.
5. According supp. Fig. S1 several classes are designated low resolution. This warrants more explanation. What was the criterion for removing these classes? According to the Fig. most of those do not look worse than the good classes. Quite large fractions were removed during the first sorting step, for the MALSU sample more than 50% of the images.
6. Supp. Fig. 1 contains only percentage of the substates for the individual sorting steps. In this way, the real size of the fractions is hidden. Absolute number of images and the relation to the full data set should be indicated.
7. The local resolution of the assembly factor densities of the various reconstructions has to be reported.
8. According to Fig. 1 and the cartoon in Fig. 6 GTPBP10 dissociates during the transition of state B to

state C. This appears to be at odds with the structural results as according to Fig. 1 none of the B states has density for GTPBP.

9. The cartoon in Fig. 6 does not depict the MALSU1 complex nor tRNA. What is the reason?

10. Table 1 is incomplete. It provides information about 4 cryo-EM maps only but 9 intermediates are described in the manuscript.

11. Some parts of several cryo-EM maps in Fig. 1 are shown at reduced contour level. This is dangerous as it gives a false impression of the strength of the signal, especially to non-experts. At least a supplemental Figure has to be shown with a comparison of the original maps and the enhanced maps.

12. Cryo-EM maps shown in a gallery should be at the same size (Fig. 1). The close ups in Fig. 5 are in a different orientation as the overall map/model.

Reviewer #4 (Remarks to the Author):

There is hardly any structural information on human mitoribosomal assembly. So far only a few structures of mitoribosome assembly intermediates from three different eukaryotic species have been solved. Some of these correspond to mitoribosomes from Trypanosomatids, which show a highly reduced rRNA content and many mitochondria-specific proteins and may not be good examples to understand mitoribosome assembly in Homo sapiens.

The authors of the current manuscript have solved several cryo-EM structures representing distinct assembly intermediates of human mitoribosomes. These findings help us to understand not only the last steps on the mitoribosome assembly pathway but also the sequence of events needed to ensure a proper folding of domains IV and V of rRNA of large subunits in all ribosomes.

Therefore there is no doubt this work represents a major advance in the field, and yet I think the manuscript still has some room for improvement. In more detail, I have two main concerns and a few minor concerns:

Major comments:

1) The authors have been able to obtain 9 different cryo-EM maps at good resolution, and this probably reflects a good handling of the data as seen in the methods section of the manuscript. However the most interesting classes from the functional point of view, classes A1 and C are the ones at lower resolution (C is at good overall resolution, but local resolution for NSUN4 and MTERF is not so great). I think these classes would benefit the most in terms of global/local resolution improvement if the data classification procedure were revisited.

I do not disagree with the classification procedure depicted in Supplementary figure 1, and on the contrary, I think I would have done it similarly since the only protein complex you knew where to locate on the ribosome was the MALSU1 complex and it makes sense that you used MALSU1 for masked classifications. However this strategy, which is particularly good to ensure most of the classes contain MALSU1, is not appropriate to enrich your classes with GTPBP10, MRM3, FactorX, DDX28, NSUN4 and MTERF4. In fact I am quite surprised that after a pull-down using GTPBP10 as bait, only about 6% of your particles contained a density for GTPBP10.

My suggestion would be to perform, after a first 3D classification (not very restrictive in removing low resolution particles), masked classifications with signal subtraction using a large mask including GTPBP10, MRM3, FactorX, DDX28 for either the MALSU1 sample and GTPBP10 or a mask with NSUN4/MTERF4 for the MALSU1 sample. Perhaps these strategies may also help to retrieve some particles allocated in "low res" classes in your MALSU1 sample dataset, that unfortunately account for 65% of the total of particles.

Another way to enrich your C and A1 classes could be to retrieve some of the micrographs you originally discarded because their resolution estimation was above 4Å-resolution since the local resolution for all these assembly factors is far above this 4Å-limit.

2) It is not very clear for non-ribosome assembly experts what the current structures represent when compared to what is known from a structural point of view about the last steps of ribosome maturation in other species like in bacteria. Given that rRNA domains IV and V also are kept immature in 23S/25S until the last steps of large subunit maturation, a fair comparison between different species could be done based only on the different folding events that occur on domains IV and V. For example, it would be interesting to know if you think you still miss intermediate states between A1 and D based on the known structures of Trypanosomas' s mitoribosome or E. coli large subunit assembly intermediates. It would be also interesting to know which assembly factors are conserved and if there are proteins playing similar functional roles without having any apparent sequence homology.

To that end you could try to summarize all this information on a large supplemental table.

Minor comments

1) Why the density for the central protuberance seems so large in classes B1-B4 when compared to that in the fully matured ribosome (classes D1 and D2)? I would expect the opposite unless there is an additional AF bound to the CP in the immature classes.

2) Lines 97, 98. I am surprised that simply by pull-downs using MALSU1 and GTPBP10 as baits, all the ribosomes on the grids correspond to immature mitoribosomes without any sign of other contaminants. Did you find any mature mitoribosome or cytoplasmic ribosomes during your data processing?

3) About the terminology of the different classes: A2 class is more similar to B classes than to A1, but I understand it cannot be included within B classes because H81 is not folded. I think it deserves to be classified in a different class, and therefore you would have 5 different classes instead of 4.

In the other hand, when you take a look at the different B classes, it is very clear they do not follow a plausible order. In fact they might not even represent physiological states but the result of E-site tRNA/MALSU1 dissociation upon grid preparation. In consequence I think numbering them is misleading and perhaps you should reconsider the terminology of the different classes. Perhaps swapping letters by numbers and vice versa may help.

4) Figure 1: It is an smart arrangement of all various maps in only one figure, but it is difficult to follow on its own without looking at the supplementary figure 1 at the very same time. Please state in the figure legend what MALSU1 and GTPBP10 mean. Moreover add the word "sample" after MALSU1 and GTPBP10.

5) You could not identify the identity of "factor X". Have you considered carrying on other kind of experiments? It is very clear that your pull-down strategy using MALSU1 and GTPBP10 as baits worked perfectly, so why not to try to do the same using MRM3 as bait? Perhaps MRM3 and factor X interact even in the absence of the mitoribosome and the complex can be identified using mass spectrometry.

6) Lines 338-341: I do not think that finding an E-site tRNA bound is enough to say that mitoribosome assembly do not follow the well-accepted concept that during the assembly all functional regions are kept immature to prevent premature association of translation factors/tRNA/mRNA; It is not rare to find E-site tRNAs bound in non-translating mature ribosomes from many species, from bacterial to eukaryotic ribosomes, and for example an E-site tRNA was found bound in the structure of the yeast mitoribosomal large subunit (Amunts et al., 2014).

7) Local resolution for most of the assembly factors is not good, preventing de novo model building for all of them. Given that for DDX28, MRM3 and GTPBP10 you had to build homology models, it would be important to know how reliable are these homology models. You used SWISS-MODEL to do so, but it would be interesting to know more details on how the program made these models.

8) Line 118: The figure does not show what is said in this sentence.

9) Line 189: The figure does not show what is said in this sentence.

10) Lines 189-192: You say that in T. brucei is found an homologous methyltransferase in an identical conformation to that of MRM3. However in T. brucei the "active" methyltransferase is LAF6, which is located in the same location as Factor X and not MRM3.

11) Line 247 and figure 5c: H81 looks quite close to me to the SAM of NSUN4. When you say it is too large distance, have you considered a possible flipping of a C in H81?

12) Line 260 and figure 5e: You say that there is a conformational change in mTERF4 towards a more closed conformation, but looking at figure 5e it seems the contrary, that is an opening of the protein to accommodate H68-71 on the cleft.

13) Please consider merging or removing entirely the last results' section since it is quite redundant with the first part of the Discussion

14) Lines 406-407: Please state the final concentration of the samples or at least the OD260 .

15) Line 415: It surprises me that there are more micrographs on the MALSU1 sample dataset than in the GTPBP10 sample dataset and in spite of that, only $\approx 275,000$ particles are included in good classes for the MALSU1 sample dataset whereas for the GTPBP10 sample dataset more than 1,320,000 particles are included in good classes.

16) Lines 457 and 458: I assume you have deposited all 9 maps and models. You should change the text accordingly.

17) Please update reference 38.

Point-by-point response to the reviewers' comments

Reviewer #2 (Remarks to the Author):

This manuscript from Roland Beckmann's group describes structures of late intermediates in assembly of the human mitochondrial large ribosomal subunit. The authors affinity-purified these particles using epitope-tagged late assembly factors, then determined the structures of these assembly intermediates at 3.1 to 4.3 Angstroms resolution by cryo-electron microscopy. They identified four principle intermediates A-D, plus five more "substates". Interestingly, they observe a phenomenon similar to what was previously observed for bacterial and yeast large subunit assembly, that the 5' and 3' domains of rRNA comprising the solvent-exposed side of the subunit are stably assembled first. Then rRNA domains IV and V complete their maturation to form the subunit interface, where the functional centers (e.g. the PTC) are located. Thus, this mode of pre-rRNA maturation seems to be conserved. The manuscript also describes the location, structure, and potential functions of six

assembly factors. The DEAD box protein DDX28 is bound to the immature CP and appears to prevent maturation of helices H80-H88. The methyltransferase MRM3 and GTPase GTPBP10 bind to helices H90-H93 and appear to delay maturation of this portion of the peptidyltransferase center. Surprisingly two intermediates contain a tRNA bound to the E site, perhaps performing a functional "test-drive". Finally, the NSUN4-TERF4 complex maintains helices H68-H71 of domain IV in an immature conformation, perhaps functioning as a final checkpoint.

I have no technical concerns about the manuscript. The results are very timely and of broad interest. However, I would like to offer several suggestions to make the manuscript more clear, and with all due respect, perhaps more interesting.

(1)Abstract: The phrase "Besides the MALSU1 complex, we identified..." seems somewhat abrupt. Readers may not yet know that the MALSU1 complex was used as a bait for purification. Thus, it seems strange to begin the sentence with this phrase. Perhaps, write instead: " Besides the MALSU1 complex used as bait for affinity purification, we identified..."

Answer:

We agree and edited the abstract as suggested for better readability.

(2)Introduction: The last paragraph of the Introduction seems a bit too dry. Rather, I suggest that the authors highlight here their most interesting discoveries, e.g. how conserved the rRNA maturation pathway is, that several assembly factors appear to function to delay folding of the rRNA, and that there is a tRNA present in the E site.

Answer:

We highlighted now our most interesting discoveries as suggested.

(3)Introduction, lines 48-50: Is the mitochondrial ribosome assembly pathway "more

complex,” or is it simply different from other assembly pathways? or both? If more complex, in what sense?

Answer:

Since the mammalian mitochondrial ribosome is very different from its bacterial ancestor (RNA-poor, protein-rich, tRNA instead of 5S rRNA) we assume a deviating assembly pathway and speculated that it could be more complex. We still think and our data support that mitochondrial ribosome assembly is somewhat different, however, in the revised text we omitted the speculation that it is more complex.

(4)Introduction, lines 54-58: This list of some of the assembly factors reads like a laundry list...why are these listed and not others? And, will nonexpert readers appreciate these specific names? Perhaps, it would be more clear to simply state the different functional classes of assembly factors that have been found, namely, RNA helicases, methyltransferases, and GTPases. This lengthy list also distracted me from initially focusing on the six assembly factors discussed in the Results.

Answer:

We agree and omitted the ‘laundry list’ as suggested.

(5)Introduction, lines 90-91, and Figure 1: Can the authors make more clear why they segregated the substate structures from the four major classes of particles? For example, are the latter more abundant? More different? Having two separate sets was initially distracting, to forecast the pathway.

Answer:

We agree that the initial presentation was not very clear. We now put all the particles in a single sequential order and only the states, which can carry an E-site tRNA are shown twice. A single sequential order was also possible, since we omitted the states lacking the MALSU1 complex. In agreement with the comments for reviewer 4, these could be the result of dissociation during sample preparation.

(6)Figure 1: As mentioned above, might the authors reorganize the structures into the order that they believe they mature? And, it might be useful to emphasize even more, somewhere in the manuscript (Discussion?), whether or not there is evidence for alternative pathways as seen in bacteria.

Answer:

As suggested, we did reorder the particles in the order of maturation. We don’t have any evidence for alternative pathways in our data when considering the maturation of the rRNA. Although we indicate in the new Fig. 5 that in addition to a sequential also a concerted maturation of remaining rRNA domain IV and V may be possible, we would not consider that as ‘alternative pathway’. We emphasize that more clearly now in the discussion.

(7)Figure 1: the bright yellow domain IV rRNA structure is attractive, but difficult to see,

especially the labels for helices...perhaps darken the yellow to mustard color?

Answer:

We changed the too bright yellow to darker yellow as suggested.

(8)Figure1: Immature or unstructured rRNA is shown in gray. This is an important distinction...perhaps use two different colors (or two shades of gray)?

Answer:

We adjusted the rRNA representation as suggested and now indicate immature rRNA in light blue and unstructured/disordered/highly flexible rRNA in grey.

(9)Figure 2: Likewise, the pink is attractive but the labels are tricky to read...darken? (The red bL35 is difficult to see...perhaps light blue?

Answer:

We adjusted the colors as suggested and now changed the pink to a darker rosé, and show bL35 in light blue, which indeed makes the Figure clearer.

(10)It is potentially interesting that the DEAD box protein DDX28 appears to physically block maturation of the CP. Is this one of the first examples for such a (potential) function for a DEAD box protein, i.e., different from the RNA or RNP remodeling function ascribed to helicase activities? Seems interesting to me. On the other hand, the authors state on line 276 that “DDX28 remodels the intermediate ...”. Has it been proven (in the referenced work) that DDX28 in fact remodels the pre-ribosome in vivo, via an RNA helicase activity? If so, this needs to be made a little more clear. Instead, or in addition, might ATP hydrolysis propel DDX28 off of the pre-ribosome to enable subsequent movement of the CP?

Answer:

We clearly see the DDX28 in its rRNA blocking binding mode and we assume that some other DEAD box helicases have a similar function, i.e. in the pre-60S (i.e. Has1). One homologous situation has indeed been described for Trypanosoma, in which a related DEAD box has been found in the similar position at the CP¹⁻³. However, it is difficult to say whether DDX28 also has more canonical remodeling activity. In the literature is evidence for the requirement of ATP hydrolysis, yet, without distinguishing between simple dissociation upon ATP hydrolysis or processive remodeling activity, as correctly pointed out by the referee. We adjusted the text to make this point clearer.

(11)Lines 145-147: the word “distorted” conveys a negative (mutant?) tone to me, i.e., not a normal part of the wildtype folding pathway...maybe just use the word “immature”?

Answer:

Agreed and edited as suggested

(12)Figure 3 and lines 179-180: The propinquity of the N terminal domain of MRM3 and

the RecA2 domain of DDX28 suggests that MRM3 might regulate or recruit DDX28. What might further support this idea?

Answer:

To our knowledge there is now further support for MRM3 recruiting of DDX28 or vice versa. Yet, we implemented this plausible idea now in our text by writing ‘...which may indicate a coordination in the recruitment of these two factors’.

(13)Line 202: Typo: Sbp1 should be Spb1. I think there may also be a bacterial relative of MRM3 that modifies the A loop: FtsJ? JBC vol 275 p. 16414 (2000).

Answer:

We thank the reviewer for pointing out the typo that we corrected accordingly. In addition, the reviewer is correct that FtsJ is modifying the A-loop in bacteria. We added that to the text, which reads now: Apparently, in bacteria a relative of MRM3, FtsJ, executes this modification, probably in a similar way.

(14)Lines 205-218, about the GTPase GTPBP10: The authors might add that, in addition to the bacterial GTPase ObgE, the yeast GTPase Nog1 also binds to and displaces helix 89.

Answer:

We added this as suggested and refer to Wu et al., 2016.

(15)Starting with line 215: It is fascinating that a tRNA is found in the E site in two apparently nonconsecutive assembly intermediates B and D. The authors do suggest that instead of maturing through state C, state B might directly transition to state D, or that either might occur. Perhaps the authors might elaborate further in the Discussion about alternative assembly pathways. In addition, the authors discuss, correctly I believe, that it is unusual to see interaction of a translation factor with a potential active site in an immature ribosome. Might the authors relate this to others’ work describing test driving of functional centers?

Answer:

As suggested, we elaborate now in the Discussion about alternative pathways, however, for which we have very little evidence. We also relate our findings on the E-site tRNA to the test driving idea by the Karbstein lab in the Discussion.

(16)Lines 264-286: this last section of the Results nicely summarizes a significant fraction of the authors’ findings in this manuscript. Therefore, might it better fit at the beginning of the Discussion? (Easy to change!)

Answer:

We agree with the reviewer and changed that accordingly.

(17)Line 273: The term “reversed order” referring to the observation that the terminal

domains of rRNA mature before the middle domains seems a little confusing to me. Could the authors use a better description?

Answer:

We agree and replaced the term 'reversed order' with 'ends-to-middle' order, which indeed describes the situation more accurately.

(18)The last section of Results nicely summarizes what was found about each late assembly factor, but omits mention of MRM3 and factor X.

Answer:

The MRM3 complex is now mentioned twice in this section of the revised text.

(19)Discussion, lines 303-306. The authors reference prior work showing that the order of rRNA accomodation that they observe in mitochondrial large subunit assembly, maturation of domains containing the two ends of the RNA before the middle portion, also occurs in bacteria and yeast. The authors should also reference another publication that demonstrated this order of maturation of domains in the large subunit: (Gamalinda et al., Genes and Development 28,198-210 (2014)).

Answer:

We totally agree with the Reviewer and added this reference that we accidentally missed.

Sincerely,
John Woolford

We are grateful for the very instructive criticism.

Reviewer #3 (Remarks to the Author):

In the present paper Chen and colleagues analyzed the assembly of the human 39S subunit of the mitoribosome. Assembly intermediates were purified from cells via a Flag tag introduced at assembly factors MALSU1 and GTPBP10, respectively, and subjected to cryo-EM. From both specimen combined the authors obtained 9 cryo-EM maps of assembly intermediates of the 39S subunit at 3.1-4.3 Å resolution informing on late assembly steps. The authors can follow transitions of the 16S rRNA involved in maturation of the central protuberance and the peptidyltransferase center. They can also identify several assembly factors although insights here are hampered by low local resolution. The results are interesting and provide new insights into the important assembly pathway of the human mitoribosomes. However, there are a couple of points the authors may want to consider to improve their manuscript.

Specific Points:

1. The results part is difficult to follow and appears cluttered. In the abstract four distinct late state intermediates are mentioned, according to the beginning of the results section “nine well defined classes which showed clearly different folding states of the 16S rRNA” were obtained and according to supp. Fig. 1 there are even more reconstructions that are not discussed (states D1, D2). It may help, to explicitly state which states were derived from which specimen and to provide an overview about the assembly factor composition of the various states, maybe in form of a table.

Answer:

We agree with the reviewer and now provide a revised new Fig. 1 and table summarizing the assembly factor composition of the various states.

2. The figure legends should be update to explain the labels. In Fig. 1 for example, the lines connecting the states are not explained, in Fig. 5 a legend for the coloring of surface potential is missing.

Answer:

We now provide a revised new Fig. 1, which now should be self-explanatory. We are sorry about the missing coloring value of surface potential, which we now added directly in the figure panel.

3. It may be also beneficial to define upfront what a state is and what a substate. States A1 and B1 contain more factors, but A2 and B3 appear to be more abundant and can thus be considered the dominating substates.

Answer:

As mentioned above, we provide a new classification of states illustrated in the new Fig. 1 and the table summarizing all factors. We hope that this makes the classification of the states more accessible.

4. From a technical point of view it is unclear, if the authors have really fully exploited the potential of method. In general the density corresponding to the assembly factors appears to be poor limiting insights into the detailed molecular interactions. The only state that appears to have density for GTPBP10 is state A1. Even in the GTPBP10 affinity purified specimen, where most of the complexes are expected to have GTPBP10, A1 is present to about 1% only (15% of 7.4%). Thus 99% of the complexes have lost the factor during preparation. It may be worthwhile to optimize the purification protocol and to explore crosslinking to improve complex stability. Also the application of specialized image processing methods such as 3D variability analysis implemented in the cryoSPARC software package may be beneficial.

Answer:

The reviewer is correct that we can see GTPBP10 only in a very small subset of states despite the fact that it was used as a bait for purification. However, the main reason for GTPBP10 not being visible in our maps is that in most states it remains flexible together with rRNA helix H89, which it binds to. To that end also chemical crosslinking cannot improve the situation since it may interconnect proteins and RNA, but does not reduce the conformational space they explore when being dynamic.

As suggested, we indeed did a 3D variability analysis in cryoSPARC, however, without improving the interpretability of the resulting maps. This would be in agreement with the very high degree of flexibility in this region as proposed by us.

5. According supp. Fig. S1 several classes are designated low resolution. This warrants more explanation. What was the criterion for removing these classes? According to the Fig. most of those do not look worse than the good classes. Quite large fractions were removed during the first sorting step, for the MALSU sample more than 50% of the images.

Answer:

This is a correct observation by the reviewer, yet, not an unusual finding in our datasets. The criterion for removing these classes was that we could neither further classify nor refine them to higher resolution. Apparently, in preparations of native complexes we often have large numbers of 'low quality' particles which cannot be used for further structural analysis. One reason may be that we are not very strict in deselecting particles after the initial 2D classification in order to not miss out on certain orientations. Therefore, we actually expect 'bad' particles (local distortions, contaminations, bad ice etc.) which we usually get rid of during 3D classification (representing the bad classes).

6. Supp. Fig. 1 contains only percentage of the substates for the individual sorting steps. In this way, the real size of the fractions is hidden. Absolute number of images and the relation to the full data set should be indicated.

Answer:

As suggested by the reviewer, we now added the absolute numbers to the substates in Supp. Fig. 1 and 2.

7. The local resolution of the assembly factor densities of the various reconstructions has to be reported.

Answer:

As suggested by the reviewer, we now added the local resolution maps of the assembly factor densities of the various reconstructions in to Supp. Fig. 5.

8. According to Fig. 1 and the cartoon in Fig. 6 GTPBP10 dissociates during the transition of state B to state C. This appears to be at odds with the structural results as according to Fig. 1 none of the B states has density for GTPBP.

Answer:

Similar as the yeast ObgE type GTPase Nog1, GTPBP10 also mainly interacts with immature H89 of the 16S rRNA. In the state 3 (former state B), H89 is highly flexible and invisible. Thus, as a binding partner, GTPBP10 is also invisible, however still flexibly associated with the particle. When further matured to state 4 (former state C), the H89 is already almost matured, thus GTPBP10 cannot bind anymore.

9. The cartoon in Fig. 6 does not depict the MALSU1 complex nor tRNA. What is the reason?

Answer:

As suggested by the reviewer, now we also added the tRNA in our Fig. 6. However, as suggested by reviewer #4, we cannot exclude that the disappearing of MALSU1 is due to the purification or grid making procedures. Thus, we removed them from our main maturation path and did not show them in the revised Fig. 6.

10. Table 1 is incomplete. It provides information about 4 cryo-EM maps only but 9 intermediates are described in the manuscript.

Answer:

The reviewer is correct and we are sorry for missing out on this information, because initially we planned to include only the most important states. Now, we added the complete information for all nine intermediates.

11. Some parts of several cryo-EM maps in Fig. 1 are shown at reduced contour level. This is dangerous as it gives a false impression of the strength of the signal, especially to non-experts. At least a supplemental Figure has to be shown with a comparison of the original maps and the enhanced maps.

Answer:

We absolutely agree with the reviewer and aim at being as transparent as possible to avoid false impressions. Therefore, we show now all maps in the revised Fig. 1 using the same contour level.

12. Cryo-EM maps shown in a gallery should be at the same size (Fig. 1). The close ups in

Fig. 5 are in a different orientation as the overall map/model.

Answer:

We agree and, as suggested by the reviewer, we provide a revised Fig. 1 in which all the cryo-EM maps were shown in the same size. We also provide a thumbnail for panel b in Fig. 5 which indicates the orientation.

Reviewer #4 (Remarks to the Author):

There is hardly any structural information on human mitoribosomal assembly. So far only a few structures of mitoribosome assembly intermediates from three different eukaryotic species have been solved. Some of these correspond to mitoribosomes from Trypanosomatids, which show a highly reduced rRNA content and many mitochondria-specific proteins and may not be good examples to understand mitoribosome assembly in *Homo sapiens*.

The authors of the current manuscript have solved several cryo-EM structures representing distinct assembly intermediates of human mitoribosomes. These findings help us to understand not only the last steps on the mitoribosome assembly pathway but also the sequence of events needed to ensure a proper folding of domains IV and V of rRNA of large subunits in all ribosomes.

Therefore there is no doubt this work represents a major advance in the field, and yet I think the manuscript still has some room for improvement. In more detail, I have two main concerns and a few minor concerns:

Major comments:

1) The authors have been able to obtain 9 different cryo-EM maps at good resolution, and this probably reflects a good handling of the data as seen in the methods section of the manuscript. However the most interesting classes from the functional point of view, classes A1 and C are the ones at lower resolution (C is at good overall resolution, but local resolution for NSUN4 and MTERF is not so great). I think these classes would benefit the most in terms of global/local resolution improvement if the data classification procedure were revisited.

I do not disagree with the classification procedure depicted in Supplementary figure 1, and on the contrary, I think I would have done it similarly since the only protein complex you knew where to locate on the ribosome was the MALSU1 complex and it makes sense that you used MALSU1 for masked classifications. However this strategy, which is particularly good to ensure most of the classes contain MALSU1, is not appropriate to enrich your classes with GTPBP10, MRM3, FactorX, DDX28, NSUN4 and MTERF4. In fact I am quite surprised that after a pull-down using GTPBP10 as bait, only about 6% of your particles contained a density for GTPBP10.

Answer:

In our 3D classification procedures, we always perform general classification without focused mask after 2D classification in the beginning. This applies to both the MALSU1 sample and GTPBP10 sample. In this way, we already could classify them to different states. The 3D classification focused on MALSU1 complex was only used to finally sub-sort state 3 to separate state 3A and 3C or 3B and 3D.

As shown in Fig. 4 and mentioned above, GTPBP10 is mainly interacting with immature rRNA H89. In all the sub-states 3, H89 is highly flexible and therefore invisible. Thus, GTPBP10 most likely moves together with it and also invisible, but it is still associated with the particles.

My suggestion would be to perform, after a first 3D classification (not very restrictive in removing low resolution particles), masked classifications with signal subtraction using a large mask including GTPBP10, MRM3, FactorX, DDX28 for either the MALSU1 sample and GTPBP10 or a mask with NSUN4/mTERF4 for the MALSU1 sample. Perhaps these strategies may also help to retrieve some particles allocated in “low res” classes in your MALSU1 sample dataset, that unfortunately account for 65% of the total of particles.

Answer:

Essentially following the suggestion by the reviewer, we tried focused classification right after 2D classification with all the particles (non-restrictive). As a result, only the focused classification on NSUN4 and mTERF4 worked, not the one focused on MRM3, GTPBP10 and others. However, the new NSUN4-mTERF4 class contains 87661 particles (our current class contains a very similar number: 83176). Unfortunately, there is no improvement on either NSUN4 or mTERF4 and the average resolution is 3.3 Å, worse than the current one. As also explained above, since our native pre-ribosomal sample is not ideal, we expect a lot of ‘bad’ particles being picked in the first step. Usually, after 2D classification, we only omit clear ice contamination and still keep low resolution classes to avoid missing out on rare orientations. However, in our experience we can discard all the bad particles during 3D classification in order to keep and enrich particles which have rare orientations. In this way, a large amount of the particles in the first round of 3D classification will be omitted due to low resolution.

In addition, we also attempted to do classification focusing on the individual factors after initial 3D classification, however, mostly without yielding any improved maps or resolution. Yet, in an approach similar to the suggested one, we combined the resulting state 1 particles of three datasets and went for a classification focused on MRM3/DDX28, thereby indeed yielding improvement of local resolution to resolve at least secondary structure. We thank the reviewer for the constructive ideas.

Another way to enrich your C and A1 classes could be to retrieve some of the micrographs you originally discarded because their resolution estimation was above 4Å-resolution since the local resolution for all these assembly factors is far above this 4Å-limit.

Answer:

We agree with the reviewer. Unfortunately, since we expected high resolution at the beginning, all the micrographs which did not contain information better than 4 Å resolution had already been automatically deleted during data collection.

2) It is not very clear for non-ribosome assembly experts what the current structures represent when compared to what is known from a structural point of view about the last steps of ribosome maturation in other species like in bacteria. Given that rRNA domains IV and V also are kept immature en 23S/25S until the last steps of large subunit maturation, a fair comparison between different species could be done based only on the different folding events that occur on domains IV and V.

For example, it would be interesting to know if you think you still miss intermediate states between A1 and D based on the known structures of Trypanosomas’s mitoribosome or E.

E. coli large subunit assembly intermediates. It would be also interesting to know which assembly factors are conserved and if there are proteins playing similar functional roles without having any apparent sequence homology.

To that end you could try to summarize all this information on a large supplemental table.

Answer:

We agree with the reviewer that an in-depth cross-species analysis (human Mito vs bacteria and *Trypanosoma* Mito) would be interesting. Yet, this is not so easy since there are only very few states available so far from other species which clearly correspond to our intermediates. The publication by the Gao group⁴ reports on later states and the Williamson analysis⁵ entirely relies on accumulation of intermediates after deleting ribosomal proteins resulting in dozens of states populating parallel assembly pathways. The publication by the Spahn group^{6,7} analyses one later native complex and *in vitro* assembled intermediates. Nevertheless, we made an attempt and, as suggested, compared our states with the corresponding bacterial ones from the Spahn publication and with the *Trypanosoma* intermediates. We now provide the comparison of different assembly states with *E. coli* in Supp. Table 3, and comparison of different assembly factors with *Trypanosoma brucei* in Supp. Table 4.

Regarding the completeness of our analysis, we expect to have observed only the most stable and populated classes of intermediates and assume that there are more intermediates between states A and D (now state 1 to 5) as proposed by the reviewer.

Minor comments

1) Why the density for the central protuberance seems so large in classes B1-B4 when compared to that in the fully matured ribosome (classes D1 and D2)? I would expect the opposite unless there is an additional AF bound to the CP in the immature classes.

Answer:

In all our states (from 1 to 5) the valine tRNA together with its associated mitoribosomal proteins were already recruited. At the given limited local resolution we could not see any differences in this region. The somewhat exaggerated size of the CP was rather caused by the choice of contour level when generating the figures, which is not easy considering the stark anisotropy of resolution. We now provide a new revised Fig. 1 displaying the CP in the best possible way, which shows that the presence of any additional factors is highly unlikely.

2) Lines 97, 98. I am surprised that simply by pull-downs using MALSU1 and GTPBP10 as baits, all the ribosomes on the grids correspond to immature mitoribosomes without any sign of other contaminants. Did you find any mature mitoribosome or cytoplasmic ribosomes during your data processing?

Answer:

We could not find mature 55S mitoribosome (apart from close to mature 39S containing MALSU1, state 5) or mature 80S ribosome.

3) About the terminology of the different classes: A2 class is more similar to B classes than to A1, but I understand it cannot be included within B classes because H81 is not folded. I think it deserves to be classified in a different class, and therefore you would have 5 different classes instead of 4.

In the other hand, when you take a look at the different B classes, it is very clear they do not follow a plausible order. In fact they might not even represent physiological states but the result of E-site tRNA/MALSU1 dissociation upon grid preparation. In consequence I think numbering them is misleading and perhaps you should reconsider the terminology of the different classes. Perhaps swapping letters by numbers and vice versa may help.

Answer:

We agree with the reviewer and revised the terminology accordingly. As suggested, we now swapped the letters and numbers, and classified state A2 as a new state (state 2). We also agree that one cannot rule out the possibility that the missing MALSU1 complex was due to the grid preparation, thus we moved the two sub states (state 3C and 3D) into Suppl. Fig. 3.

4) Figure 1: It is an smart arrangement of all various maps in only one figure, but it is difficult to follow on its own without looking at the supplementary figure 1 at the very same time. Please state in the figure legend what MALSU1 and GTPBP10 mean. Moreover add the word “sample” after MALSU1 and GTPBP10.

Answer:

We agree with the reviewer and provide now a new revised Fig. 1. At the same time, we also provide a new Suppl. Table 2 to indicate the source of different states.

5) You could not identify the identity of “factor X”. Have you considered carrying on other kind of experiments? It is very clear that your pull-down strategy using MALSU1 and GTPBP10 as baits worked perfectly, so why not to try to do the same using MRM3 as bait? Perhaps MRM3 and factor X interact even in the absence of the mitoribosome and the complex can be identified using mass spectrometry.

Answer:

With the help of more data and further classification, in our new state 1, we found the density of Factor X to be highly similar to MRM3 (as far as the still limited resolution allows). At the same time, the *Trypanosoma brucei* mt-LAF6, which binds to the same position as Factor X, is the active methyltransferase. This indicates that mt-LAF6 is the direct homolog of MRM3. Also considering that SpoU-type of methyltransferases always form homodimers, we now tentatively assign Factor X as another copy of MRM3, which needs to be confirmed at higher resolution in future studies.

6) Lines 338-341: I do not think that finding an E-site tRNA bound is enough to say that mitoribosome assembly do not follow the well-accepted concept that during the assembly all functional regions are kept immature to prevent premature association of translation factors/tRNA/mRNA; It is not rare to find E-site tRNAs bound in non-translating mature ribosomes from many species, from bacterial to eukaryotic ribosomes, and for example an

E-site tRNA was found bound in the structure of the yeast mitoribosomal large subunit (Amunts et al., 2014).

Answer:

We agree that tRNA has often been found bound to the E-site of mature cytoplasmic and mitochondrial ribosomes as pointed out by the reviewer. However, this is the first time to observe a translation component that so far is only described to bind mature ribosomes has been found on an immature assembly intermediate. To that end we dare to disagree with the reviewer and consider it justified to state that mitoribosome assembly is not following the concept of avoiding premature translation factor/tRNA/mRNA binding.

7) Local resolution for most of the assembly factors is not good, preventing de novo model building for all of them. Given that for DDX28, MRM3 and GTPBP10 you had to build homology models, it would be important to know how reliable are these homology models. You used SWISS-MODEL to do so, but it would be interesting to know more details on how the program made these models.

Answer:

As suggested, we now added the information on the homologous structures we used as templates when employing SWISS-MODEL in our methods part, and refer to the publication describing the underlying technology of SWISS-MODEL for more details.

8) Line 118: The figure does not show what is said in this sentence.

Answer:

H81 as well as bL33 which are mentioned in former Line 118 are color coded and labelled now in the revised Fig. 1.

9) Line 189: The figure does not show what is said in this sentence.

Answer:

We agree and corrected that mistake in the revised text by referring to Fig. 3 b

10) Lines 189-192: You say that in *T. brucei* is found an homologous methyltransferase in an identical conformation to that of MRM3. However in *T. brucei* the “active” methyltransferase is LAF6, which is located in the same location as Factor X and not MRM3.

Answer:

As explained above in more detail, based on a better resolved map, we now tentatively assign the Factor X density to an active second copy of MRM3, which would be in perfect agreement with the findings in *T. brucei*.

11) Line 247 and figure 5c: H81 looks quite close to me to the SAM of NSUN4. When you say it is too large distance, have you considered a possible flipping of a C in H81?

Answer:

We now provide a new panel which was shown in a slightly different orientation which better illustrates the actual distance. We also label the distance (17 Å) which is too large, even we considering a possible flipping out of the base to be modified. Yet, we have good enough resolution for H81 of 16S rRNA, which clearly shows that it is not flipped out.

12) Line 260 and figure 5e: You say that there is a conformational change in mTERF4 towards a more closed conformation, but looking at figure 5e it seems the contrary, that is an opening of the protein to accommodate H68-71 on the cleft. unambiguously

Answer:

We absolutely agree with the reviewer and are sorry for this mistake. We corrected it accordingly.

13) Please consider merging or removing entirely the last results' section since it is quite redundant with the first part of the Discussion

Answer:

As suggested by the reviewer, we merged/moved the last results section into the Discussion.

14) Lines 406-407: Please state the final concentration of the samples or at least the OD260

.

Answer:

As suggested by the reviewer, now we provide the final OD260 in the method section.

15) Line 415: It surprises me that there are more micrographs on the MALSU1 sample dataset than in the GTPBP10 sample dataset and in spite of that, only $\approx 275,000$ particles are included in good classes for the MALSU1 sample dataset whereas for the GTPBP10 sample dataset more than 1,320,000 particles are included in good classes.

Answer:

This apparent mis-proportion may simply be a result the MALSU1 sample being less concentrated (approx. half the OD260 of the GTPBP10 sample) and the data of the MALSU1 sample are of inferior quality containing more contamination.

16) Lines 457 and 458: I assume you have deposited all 9 maps and models. You should change the text accordingly.

Answer:

We now indeed deposited all 9 maps and change the text accordingly.

17) Please update reference 38.

Answer:

We updated this reference as suggested.

- 1 Jaskolowski, M. *et al.* Structural Insights into the Mechanism of Mitoribosomal Large Subunit Biogenesis. *Molecular cell* **79**, 629-644 e624, doi:10.1016/j.molcel.2020.06.030 (2020).
- 2 Soufari, H. *et al.* Structure of the mature kinetoplastids mitoribosome and insights into its large subunit biogenesis. *Proceedings of the National Academy of Sciences of the United States of America* **117**, 29851-29861, doi:10.1073/pnas.2011301117 (2020).
- 3 Tobiasson, V. *et al.* Interconnected assembly factors regulate the biogenesis of mitoribosomal large subunit. *The EMBO journal* **40**, e106292, doi:10.15252/embj.2020106292 (2021).
- 4 Li, N. *et al.* Cryo-EM structures of the late-stage assembly intermediates of the bacterial 50S ribosomal subunit. *Nucleic acids research* **41**, 7073-7083, doi:10.1093/nar/gkt423 (2013).
- 5 Davis, J. H. *et al.* Modular Assembly of the Bacterial Large Ribosomal Subunit. *Cell* **167**, 1610-1622 e1615, doi:10.1016/j.cell.2016.11.020 (2016).
- 6 Nikolay, R. *et al.* Structural Visualization of the Formation and Activation of the 50S Ribosomal Subunit during In Vitro Reconstitution. *Molecular cell* **70**, 881-893 e883, doi:10.1016/j.molcel.2018.05.003 (2018).
- 7 Nikolay, R. *et al.* Snapshots of native pre-50S ribosomes reveal a biogenesis factor network and evolutionary specialization. *Molecular cell* **81**, 1200-1215 e1209, doi:10.1016/j.molcel.2021.02.006 (2021).

REVIEWERS' COMMENTS

Reviewer #2 (Remarks to the Author):

The authors have satisfied all of my requests to improve the ms. It also appears to me that they have done so for the other two reviewers.

Reviewer #3 (Remarks to the Author):

Chen and colleagues have thoroughly revised and improved their manuscript. However, there are still some points that warrant attention.

1. The beginning of the results part still lacks clarity. On page 7, line 101 the authors write that they obtained nine well defined classes, which showed clearly different folding states of the 16S rRNA. However, a little later (line 112) they state that there are only five principle states according to the folding of the 16S rRNA and that further sub-states are due to different AF composition. In Fig. 1 they only show seven states. Why? They refer to Fig. 1 for state 3D (page8, line 131) but state 3D is not shown in Fig. 1.
2. In the answer to reviewer #4 the authors quote the recent paper by Nikolay et al., 2021, which however is not referenced in the paper. As it seems highly relevant, it should be properly quoted and discussed.
3. The presence/absence of density for GTPBP10 and the timing of its release (point 8 of reviewer #3, point 1 of reviewer #4) is not discussed sufficiently. In the answers to the reviewers the authors suggest that GTPBP10 interact with immature H89 and is therefore invisible. While this may be, a proper discussion of this important issue is lacking in the manuscript. Is there any evidence for this model and can it be ruled out that GTPBP10 has been lost after affinity purification? Furthermore, it seems strange then that GTPBP is visible in state 1. Is H89 more mature in state 1 than in state 3?

Reviewer #4 (Remarks to the Author):

I acknowledge that the authors took into consideration most of my remarks from the initial review. I do not have further concerns at this time.

Point-by-Point response

Reviewer #2 (Remarks to the Author):

The authors have satisfied all of my requests to improve the ms. It also appears to me that they have done so for the other two reviewers.

Answer:

We thank the reviewer for the constructive comments.

Reviewer #3 (Remarks to the Author):

Chen and colleagues have thoroughly revised and improved their manuscript. However, there are a still some points that warrant attention.

Answer:

We thank the reviewer for the positive comments.

1. The beginning of the results part still lacks clarity. On page 7, line 101 the authors write that they obtained nine well defined classes, which showed clearly different folding states of the 16S rRNA. However, a little later (line 112) they state that there are only five principle states according to the folding of the 16S rRNA and that further sub-states are due to different AF composition. In Fig. 1 they only show seven states. Why? They refer to Fig. 1 for state 3D (page8, line 131) but state 3D is not shown in Fig. 1.

Answer:

We are sorry for the mistake and lack of clarity. Now, we revised the sentence in the beginning of the results as follows 'After processing, we obtained nine well defined classes which represent five different folding states of the 16S rRNA with some additional variation regarding AF or tRNA association (Fig. 1, Supplementary Fig. 1, Supplementary Fig. 2 and Supplementary Fig. 3).'

We indeed could find nine defined classes from our dataset based on differences in rRNA conformation, tRNA association and AF composition. However, another reviewer suggested that the lack of the MALSU1 complex could simply be a result of dissociation during the purification procedure. We agree that we cannot exclude this possibility and therefore consider two states that occur in the presence and absence of the MALSU1 complex essentially the same. In our revised manuscript we therefore moved two classes (state 3C and 3D) into supplementary Fig. 3 and as a result, only seven main classes/intermediates were shown in Fig. 1.

Again, we are sorry for the lack of clarity. In our revised manuscript, for the illustration of the 9 new states we now refer to both, Fig. 1 and Supplementary Fig. 3. We refer to Supplementary Fig. 3 alone for description of state 3D. The new text reads now:

" In detail, state 3A contained both the MALSU1 complex and the E-site tRNA, whereas state 3B and 3C contained either the MALSU1 complex or tRNA (Fig. 1 and Supplementary Fig. 3). Surprisingly state 3D contained neither one (Supplementary Fig. 3)."

2. In the answer to reviewer #4 the authors quote the recent paper by Nikolay et al., 2021, which however is not referenced in the paper. As it seems highly relevant, it should be properly quoted and discussed.

Answer:

We agree and we quote now the paper by Nikolay et al., 2021, in our revised manuscript and discuss it shortly in the main text.

3. The presence/absence of density for GTPBP10 and the timing of its release (point 8 of reviewer #3, point 1 of reviewer #4) is not discussed sufficiently. In the answers to the reviewers the authors suggest that GTPBP10 interact with immature H89 and is therefore invisible. While this may be, a proper discussion of this important issue is lacking in the manuscript. Is there any evidence for this model and can it be ruled out that GTPBP10 has been lost after affinity purification? Furthermore, it seems strange then that GTPBP is visible in state 1. Is H89 more mature in state 1 than in state 3?

Answer:

We now added a new discussion on GTPBP10. "In this state (state 3), although we could not rule out that GTPBP10 might be dissociated during preparation, GTPBP10 is expected to still be associated with the flexible and therefore invisible 16S rRNA helix H89 since the same state was obtained using different approaches. Thus, GTPBP10 most likely moves together with H89 without providing visible density, but it is still associated with the particles."

We cannot rule out the possibility that GTPBP10 could dissociate during affinity purification. However, since using a different affinity tag (MALSU1) and even without affinity purification (Brown A, et al, NSMB, 2017), we could find the same states which show the same disordered conformation of the H89 region, it is most plausible that GTPBP10 is flexible and thus invisible together with H89. At the given resolution, since H89 is immature in both states 1 and 3, we cannot derive any conclusions regarding a possible difference in maturation of this region between states 1 and 3.

Reviewer #4 (Remarks to the Author):

I acknowledge that the authors took into consideration most of my remarks from the initial review.

I do not have further concerns at this time.

Answer:

We thank the reviewer for the constructive comments.